# Beryl Mineralogy and Fluid Inclusion Constraints on the Be Enrichment in the Dakalasu No.1 Pegmatite, Altai, NW China

**Qingyu Suo** [1,2,3], **Ping Shen** [1,2,3,*], **Yaoqing Luo** [1,2,3], **Changhao Li** [1,3], **Haoxuan Feng** [1,3], **Chong Cao** [1,3,4], **Hongdi Pan** [5] **and Yingxiong Bai** [1,2,3]

[1] Key Laboratory of Mineral Resources, Institute of Geology and Geophysics, Chinese Academy of Sciences, Beijing 100029, China; suoqingyu20@mails.ucas.ac.cn (Q.S.); yqluo@mail.iggcas.ac.cn (Y.L.); lch61522191@mail.iggcas.ac.cn (C.L.); fenghaoxuan15@mails.ucas.ac.cn (H.F.); caochong1016@163.com (C.C.); baiyx@mail.iggcas.ac.cn (Y.B.)

[2] Innovation Academy for Earth Science, Chinese Academy of Sciences, Beijing 100029, China

[3] College of Earth and Planetary Sciences, University of Chinese Academy of Sciences, Beijing 100049, China

[4] College of Mining Engineering, North China University of Science and Technology, Tangshan 063210, China

[5] School of Earth Science and Resources, Chang'an University, Xi'an 710054, China; hongzhushi@sina.com

* Correspondence: pshen@mail.iggcas.ac.cn; Tel.: +86-010-82998189

**Abstract:** The Dakalasu No.1 pegmatitic rare-element deposit is a representative of Be-Nb-Ta pegmatites in Altai, Xinjiang, China. Beryl is the most important beryllium-carrying mineral in Dakalasu No.1 pegmatite. To constrain the concentration mechanism of Be, we conducted a study of the textural relationships and chemical compositions (major and trace elements) of beryl, along with microthermometry and Raman spectroscopy on beryl-hosted fluid inclusions. Two generations of beryl were recognized. The early beryl I was formed in the magmatic stage, whereas the late beryl IIa and IIb were formed in the magmatic-hydrothermal stage. Lithium and Cs contents increased from beryl I, beryl IIa, to beryl IIb, whereas Mg and Rb contents decreased. Scandium, V, and Ga contents of beryl IIa are similar to beryl IIb, but different in beryl I. Titanium is enriched in beryl IIa. The high FeO contents and Na/Cs ratios of beryl (I, IIa, and IIb) reveal the low degree of differentiation evolution of the Dakalasu No.1 pegmatite. Two types of melt inclusions and four types of fluid inclusions were identified in beryl IIa, IIb, and associated quartz. The microthermometry results indicated that beryl II is formed at 500 °C–700 °C, and 200 MPa–300 MPa. The Dakalasu No.1 pegmatite melt is enriched in volatiles, such as B, F, and $CO_2$, evidenced by a large amount of tourmaline in the wall zone, the occurrence of a variety of tiny cryolite ($Na_3AlF_6$) inclusions, and $CO_2$-rich fluid inclusions in beryl IIa. The enrichment mechanism of Be may be related to the crystallization of beryl at highly undercooled states of melt, and melt–melt–fluid immiscibility during the evolution and differentiation of the melt.

**Keywords:** beryl; undercooling; melt–melt–fluid immiscibility; Dakalasu No.1 pegmatite; Chinese Altai

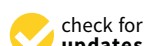



## 1. Introduction

Beryllium is an important rare element and has been widely used in aviation, aerospace, nuclear reactors, and other fields. Beryl is the most common Be-bearing mineral and is widely distributed in rare metal granites or granitic pegmatites. Beryl is hexagonal, and its ideal chemical formula is $Be_3Al_2[Si_6O_{18}]$. Beryl has a ring structure consisting of Si tetrahedra that are linked by Be tetrahedra and Al octahedra [1]. However, various other elements may be incorporated into the crystal lattice, resulting in structural distortion. For instance, lithium may substitute Be at the tetrahedral site, while trivalent (e.g., $Fe^{3+}$, $Sc^{3+}$, $Cr^{3+}$ or $V^{3+}$) and/or divalent ($Fe^{2+}$, $Mg^{2+}$ or $Mn^{2+}$) cations may substitute Al at the octahedral site. During substitution, charge balance is maintained through the incorporation of monovalent cations ($Na^+$, $Rb^+$, $K^+$ or $Cs^+$) at the channel sites [2,3]. As beryl has various forms of ion substitution, it has an important role as a geochemical tracer. Černý et al. [3]

reported that $K_2O + Na_2O + Li_2O + Rb_2O + Cs_2O(R_2O)$ of beryl in Be-Nb-Ta type pegmatite was 0.5–1.0 wt%. In pegmatite with lithium minerals, beryl is characterized by high Na, and Al contents and low Cs contents. When Cs minerals appear in pegmatite, the $R_2O$ content of beryl is generally above 2 wt%, especially if the content of $Cs_2O$ is high [3]. The variation of beryl's Na/Cs and Mg/Fe ratios can trace the degree of differentiation and the magmatic evolution of pegmatite [4–9]. The variation in trace elements (Na, K, Li, Rb, Cs, Fe) in beryl can reflect the physical and chemical conditions of its formation [10]. The melt–fluid inclusions in beryl can provide fluid information during mineralization [11,12].

The Dakalasu No.1 granitic pegmatite is one of the few Be-Nb-Ta type pegmatites in Altai, Xinjiang. There are only five Be-Nb-Ta type pegmatite deposits (Amusitai, Husite, Qukuer, Jiamukai, and Dakalasu) in the area. Previous work focused on the geochronology of the Dakalasu No.1 pegmatite, and Qin et al. [13] obtained a zircon U–Pb age of $231.8 \pm 7.4$ Ma (2σ) for the Dakalasu No.1 pegmatite and Wang et al. [14] reported a muscovite $^{40}Ar/^{39}Ar$ plateau age of $248.4 \pm 2.1$ Ma for the wall zone of the same pegmatite. Recently, a columbite sample from the wall zone of No.1 pegmatite was dated using LA-ICP-MS, yielding a U–Pb Concordia age of $239.6 \pm 3.8$ Ma (2σ) [15]. Zhou et al. [15] proposed that their obtained columbite U–Pb age represented the emplacement age of the pegmatite. Two columbite samples from the intermediate zone were also dated, displaying concordant U–Pb ages of $228.1 \pm 0.6$ Ma and $229.0 \pm 1.0$ Ma [16]. Zou and Li [17] described the occurrence and mineral assemblage of beryl in the Dakalasu No.1 pegmatite and found two types of beryl. However, a systematic analysis of the chemical composition for both the types of beryl and fluid inclusions has not been conducted.

In this paper, the texture and chemical composition of beryl in the Dakalasu No.1 pegmatite are systematically studied by using SEM-BSE, EMPA, and LA-ICP-MS analysis. In addition, beryl-hosted fluid inclusions also are studied by Raman spectroscopy and microthermometry. Based on these new data, the forming environment and evolution process of beryl are inferred.

## 2. Geological Setting

The Chinese Altai orogenic belt (CAOB) is located in northern Xinjiang, adjacent to eastern Kazakhstan, southern Russia, and western Mongolia (Figure 1a) [18–21]. This orogen is regarded as one of the most important rare-metal metallogenic belts in China [17]. According to the stratigraphy, magmatism, metamorphism, and deformation pattern of the Chinese Altai orogenic belt, from the north to south, the orogenic belt is divided into five terranes, namely, the North Altai, Northwest Altai, Central Altai, Qiongkuer-Abagong, and South Altai [19,22].

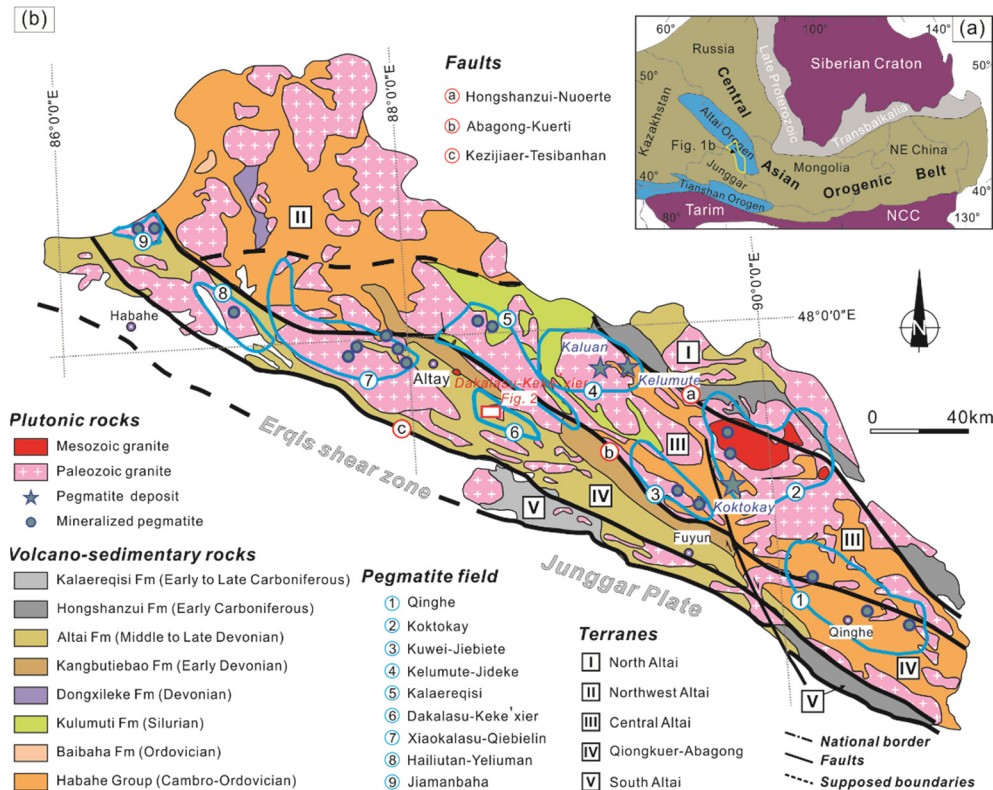

**Figure 1.** (**a**) Main tectonic units (modified after [23]), of the Chinese Altai orogen and adjacent areas; (**b**) Geological map of the Chinese Altai, showing the location of mineralized pegmatites and deposits (modified from [24]).

The Chinese Altai is a world-renowned pegmatite district. The pegmatites are mainly found in the Central Altai and Qiongkuer–Abagong terranes [25], and are divided into nine pegmatite fields according to their differing spatial distributions and types of rare-metal mineralization (Figure 1b) The Qiongkuer-Abagong terrane where the Dakalasu pegmatite field is located, mainly consists of upper Silurian–Lower Devonian arc andesitic volcanic and volcanic clastic rocks with lesser basaltic rocks [21,26]. There are more than 1000 pegmatite dikes in the Qiongkuer-Abagong terrane, and they are mainly within or around the porphyritic biotite granites ranging from Late Permian to Early Triassic at Dakalasu and two-mica granite at Kekexi'er [27,28]. The mineralization types include muscovite, muscovite-rare-element, and rare-element [17]. The Dakalasu-Kekexi'er pegmatite field belongs to the rare-element type, which is located in the Qiongkuer-Abagong terrane (Figure 1b), and is bordered by the Abagong fault to the north and the Kalasu fault to the south (Figure 2).

The Dakalasu No.1 pegmatite is a representative of the pegmatite Be-Nb-Ta deposit in this area [17]. The previous exploration reported that the Dakalasu No.1 pegmatite is a small to medium-sized deposit, with 126.9 t of BeO and 51.6 t of (Ta, Nb)$_2$O$_5$ [17]. The Dakalasu No.1 pegmatite is emplaced in the Dakalasu biotite granite (Figure 3a). According to previous studies [14,15,17], the Dakalasu No.1 pegmatite can be divided into four zones, from contact with the Dakalasu biotite granite inwards: the wall zone, intermediate zone, replacement zone, and core zone. Different symbols of quartz and microline refer to crystal grain size and color (Figure 4). The wall zone is composed of from medium- to coarse-grained quartz, feldspar, mica, tourmaline, beryl, and fine-grained garnet. Tourmaline is arranged along the contact with the biotite granite (Figure 3b). Garnet is developed in the transition area between the wall zone and intermediate zone (Figure 3c). The intermediate zone is composed of blocky feldspar and quartz, megacrystic muscovite, prismatic beryl, and columbite group minerals (CGM). There is a replacement zone between

the intermediate zone and core zone, which is composed of albite aggregates [15]. This zone may be formed by the self-metasomatism of exsolution fluids. The core zone is mainly composed of massive micro-plagioclase and quartz.

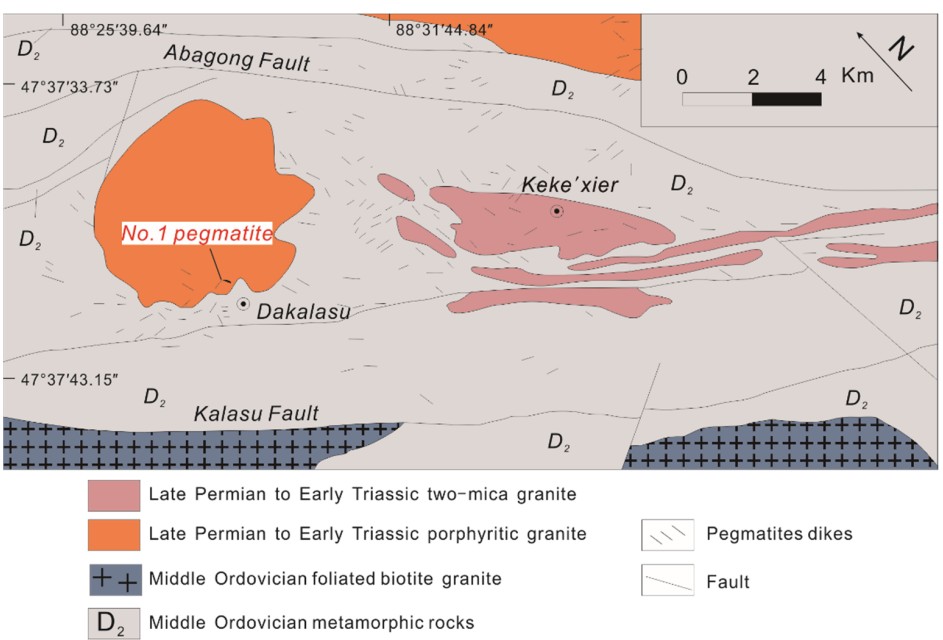

**Figure 2.** Schematic Geological Map of the Dakalasu-Kekexi'er pegmatite field, modified after [16,17].

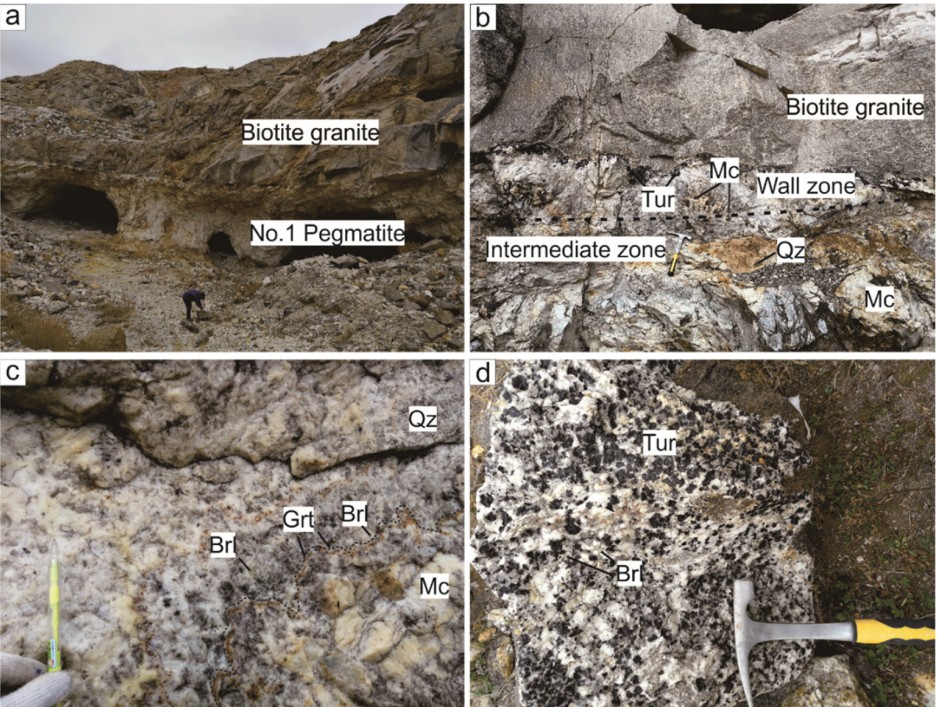

**Figure 3.** Field photographs of (**a**) The Dakalasu No.1 pegmatitic dike emplaced into the Dakalasu biotite granite; (**b**) The wall zone is dominated by coarse-grained quartz, microcline, tourmaline, beryl, and fine- to medium-grained garnet, and the intermediated zone bears blocky microline and quartz; (**c**) Garnet and beryl occur in contact with the intermediate zone along the wall zone; (**d**) Beryl coexisting with abundant tourmaline (black), microcline (white) and quartz (gray). Abbreviations: Brl = beryl, Grt = garnet, Mc = microcline, Ms = muscovite, Qz = quartz and Tur = tourmaline. Hammer = 30 cm, pen = 13.5 cm.

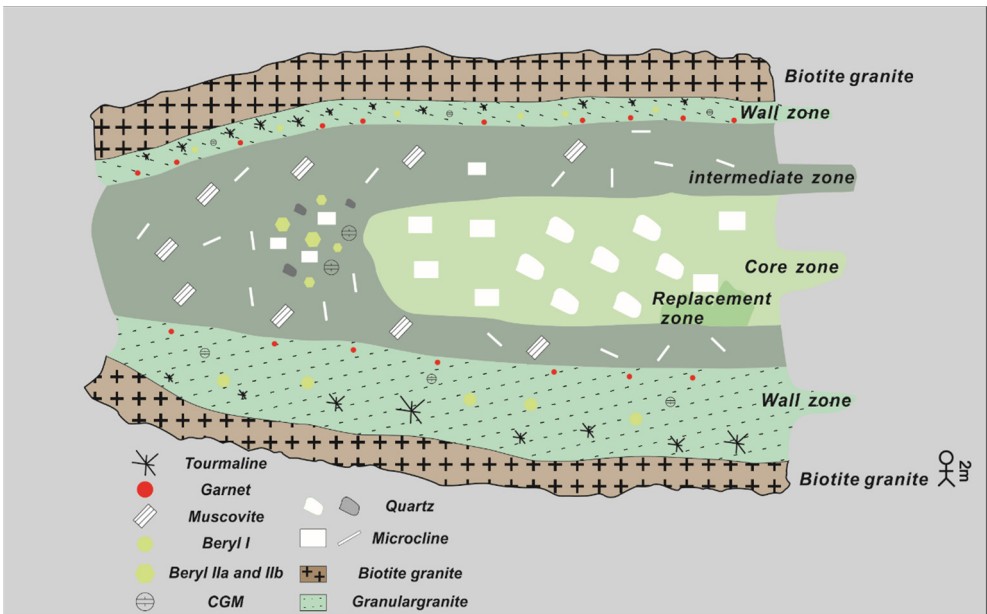

**Figure 4.** Schematic diagram of the Dakalasu No.1 pegmatite.

## 3. Samples and Analytical Methods

### 3.1. Samples

Samples ranging from medium to coarse beryl (beryl I) from the wall zone are located at the border of the Dakalasu No.1 pegmatite, close to the contact with the Dakalasu biotite granite. The prismatic beryl (beryl IIa and IIb) samples come from the intermediate zone.

Beryl I is subhedral, with a grain size of 0.5~1 cm (Figures 5a and 6a) and pale green to green in color. Texturally, it is associated with medium- to coarse-grained quartz, microcline, coarse-grained black tourmaline, fine-grained reddish garnet, and fine- to medium-grained columbite-group minerals (CGM) (Figure 3c). The beryl II crystals are characterized by several centimeters (up to 50 cm) in length and up to 30 cm in diameter, with prismatic or doubly terminated hexagonal biconical faces. They show yellow–green color and are associated with shallow gray quartz, blocky microcline, pegmatitic muscovite, and coarse CGM (Figure 5b–d). Prismatic beryl crystals often contain mineral inclusions, such as albite (Figure 6b). As shown in the BSE image, Beryl II can be divided into IIa and IIb. Compared to the beryl IIa, the beryl IIb contains fissures, which are filled with hydrothermal muscovite (Figure 6c). Both beryl IIa and IIb in thin sections are unzoned and show similar BSE images.

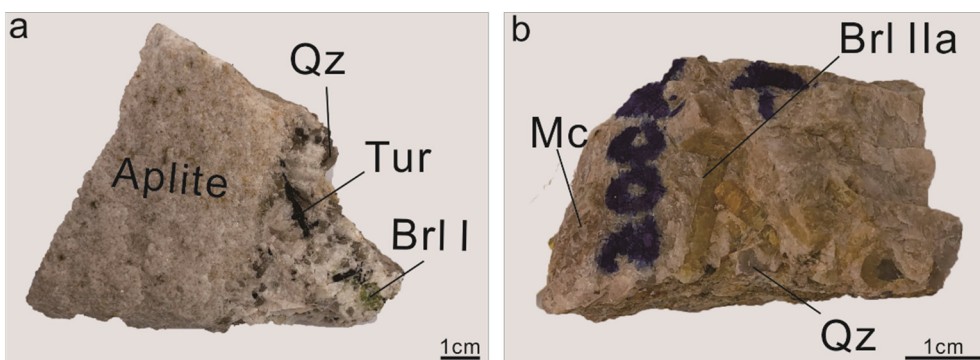

**Figure 5.** *Cont.*

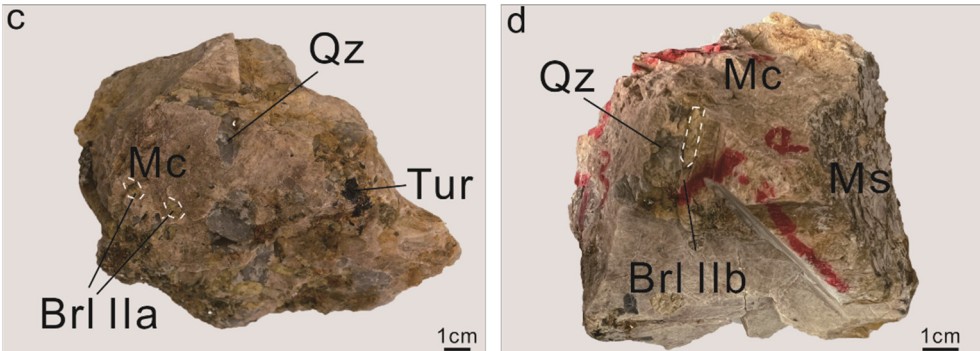

**Figure 5.** Hand specimen photographs of representative samples from different zones of the Dakalasu No.1 pegmatite. (**a**) Typically, the wall zone of granitic pegmatite, which is an intergrowth of Tur + Qz + Brl; (**b–d**) Prismatic beryl from the intermediate zone, and is associated with Qz + Mc ± Ms. Abbreviations: Brl = beryl, Qz = quartz, Mc = microcline, Ms =muscovite and Tur = tourmaline.

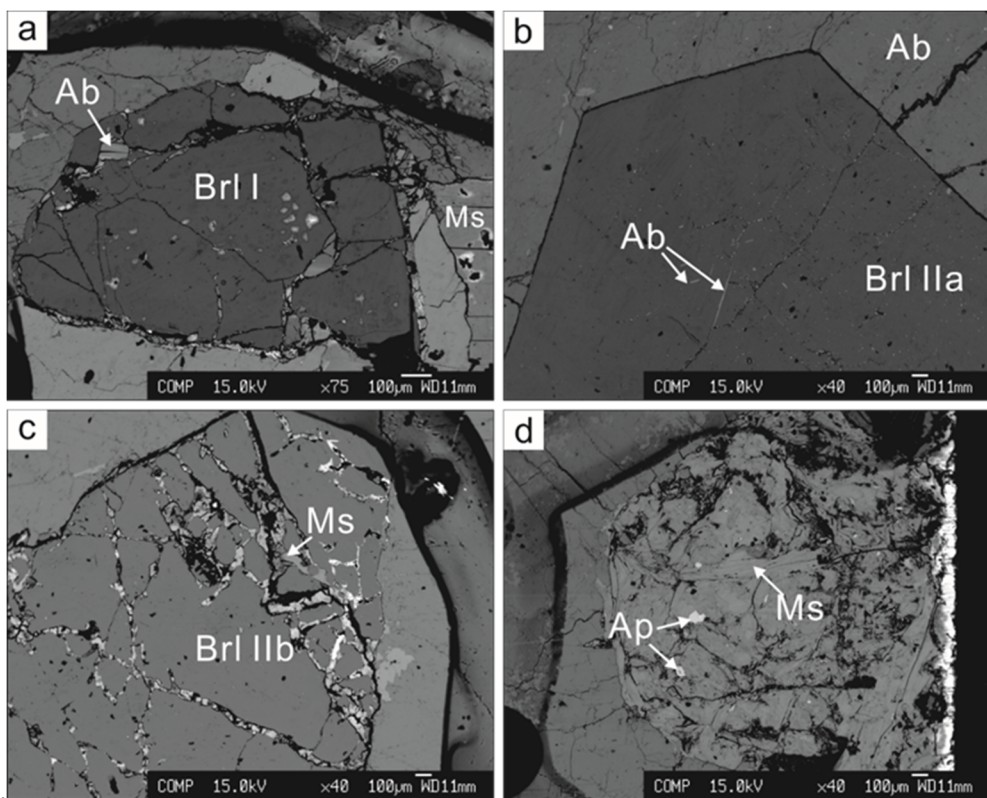

**Figure 6.** Backscattered electron (BSE) photomicrographs of beryl (Brl) from the Dakalasu No.1 pegmatite (**a**) Beryl I coexists with albite and mica overgrown, and filling the fissures in the beryl I; (**b**) Beryl IIa is closely associated with albite and contains albite crystals inside; (**c**) Beryl IIb fissures are developed, and the fissures are filled with mica and apatite; (**d**) An aggregate composed of mica and a small amount of apatite was found to metasomatized the original minerals near Beryl. Brl I: Derived from coarse-grained Beryl in the wall zone, Brl IIa, and IIb: Derived from prismatic Beryl in the intermediate zone. Abbreviations: Ab = ablite, Ap = apatite, Brl = beryl, and Ms = muscovite.

## 3.2. Scanning Electron Microscope and Backscattered Electrons (SEM-BSE)

BSE images of beryl crystals were obtained using a Nova NanoSEM 450 field emission scanning electron microscope (FE-SEM) at the Institute of Geology and Geophysics, Chinese Academy of Sciences (IGGCAS). Working conditions of the BSE imaging include 15 kV accelerating voltage and 20 nA beam current.

### 3.3. Electron-Microprobe Analysis (EMPA)

Major element compositions of beryl were measured using a JEOL JXA-8100 and CAMECA SXFive electron microprobe operated in wavelength dispersive spectrometer mode at IGGCAS, respectively. EPMA analysis of beryl was run under an accelerating voltage of 15 kV, a beam current of 20 nA with beam diameters of 2 μm, and a 10–30 s peak counting time. Natural minerals and synthetic oxides were used as standards. The raw data were corrected using the ZAF procedure. The contents of Be and Li for beryl were estimated by stoichiometry assuming the equations [29]. Representative major element compositions of beryl are summarized in Table 1.

**Table 1.** The representative EMPA results of beryl in the Dakalasu No.1 pegmatite.

| Sample N | Beryl I | Beryl IIa | Beryl IIb |
|---|---|---|---|
| | 20DKP13 (11) | 20DK1-4 (26) | 20DKLS1-1 (17) |
| $SiO_2$ wt% | 67.18 | 64.82 | 64.78 |
| $Al_2O_3$ | 16.60 | 16.96 | 16.71 |
| FeO | 0.76 | 0.44 | 0.49 |
| MnO | 0.03 | 0.01 | 0.01 |
| MgO | 0.16 | 0.05 | 0.03 |
| CaO | 0.01 | 0.01 | 0.01 |
| SrO | 0 | 0.22 | 0.23 |
| BeO* | 12.51 | 13.08 | 13.15 |
| $Li_2O$* | 0.09 | 0.16 | 0.09 |
| $Na_2O$ | 1.82 | 0.48 | 0.43 |
| $K_2O$ | 0.03 | 0.12 | 0.02 |
| $Rb_2O$ | 0.13 | 0.13 | 0.13 |
| $Cs_2O$ | 0.03 | 0.11 | 0.11 |
| Total | 99.74 | 96.58 | 96.21 |
| Formulae based on 18 oxygen atoms | | | |
| Si apfu | 6.100 | 6.07 | 6.08 |
| Al | 1.778 | 1.86 | 1.84 |
| Fe | 0.057 | 0.03 | 0.04 |
| Mn | 0.002 | 0.00 | 0.00 |
| Mg | 0.022 | 0.01 | 0.00 |
| $\Sigma$ | 1.859 | 1.904 | 1.891 |
| Ca | 0.001 | 0.00 | 0.00 |
| Be | 2.75 | 2.94 | 2.96 |
| Li | 0.25 | 0.06 | 0.04 |
| Na | 0.32 | 0.09 | 0.08 |
| K | 0.003 | 0.01 | 0.00 |
| Rb | 0.008 | 0.01 | 0.01 |
| Cs | 0.004 | 0.00 | 0.00 |
| □ | 0.905 | 0.87 | 0.89 |
| P | 0.002 | 0.00 | 0.00 |
| Na/Cs | 79.61 | 4.05 | 3.44 |

Note: bdl, below detection limit; □, vacancy; Structural formulae were calculated based on O = 18 apfu; BeO* and $Li_2O$* were estimated by stoichiometry assuming Be = 3—Li and Li = Na + K + Rb + Cs—(Fe + Mn + Mg + Zn + Ca) cited from [29].

### 3.4. LA-ICP-MS Trace Element Analysis

Trace element contents of beryl were determined by laser ablation-inductively coupled plasma-mass spectrometry (LA-ICP-MS), employing an Agilent 7500a Q-ICP-MS instrument (Agilent Technologies, the United States of America) coupled to a 193 nm ArF excimer laser system (Geolas HD, Lambda Physik, Göttingen, Germany) at the State Key Laboratory of Lithospheric Evolution, IGGCAS. The approach is similar to those outlined in Wu et al. [30] with isotopes measured using a peak-hopping mode with a laser beam diameter of ca. 44 μm. The laser energy density is ~4.0 J/cm². Helium was employed as the ablation gas to improve the transporting efficiency of ablated aerosols. NIST SRM

610 reference glass was used as a calibration material, and ARM-1 [31], and BCR-2G were analyzed for data quality control. Silicon ($^{29}Si$) was used as an internal standard. The resulting data were reduced based on the GLITTER program [32]. For most trace elements (>0.10 mg/g), the accuracy was better than ±10% and relative standard deviation (1RSD) of ±10%. Representative trace element compositions of beryl are summarized in Table 2.

**Table 2.** The LA-ICP-MS trace element compositions of beryl in the Dakalasu No.1 pegmatite.

| Sample N | Grained Beryl (I) | Prismatic Beryl (IIa) | Prismatic Beryl (IIb) |
|---|---|---|---|
| | 20DKP13 (9) | 20DK1-4(14) | 20DKLS1-1-2(12) |
| Li.ppm | 400 | 534 | 501 |
| Cs | 815 | 1476 | 1595 |
| Mg | 1200 | 366 | 186 |
| Ti | 84 | 107 | 90 |
| K | 312 | 80 | 60 |
| Sc | 1.1 | 8 | 7 |
| V | 2.3 | 4.3 | 4.9 |
| Mn | 179 | 36 | 40 |
| Ga | 22 | 63 | 62 |
| Rb | 140 | 28 | 30 |

*3.5. Raman Spectroscopy, Incluison Petrography, and Microthermometry*

A Witec alpha300R confocal-Raman spectrometer equipped with a 532.2 nm laser and diffraction gratings of 300 grooves mm$^{-1}$ (resolution is 4.8 cm$^{-1}$), and Zeiss microscope 100× objective lens (NA = 0.9), the, was used to identify volatile and solid phases in inclusions at the IGGCAS. The identifications were based on the Raman spectra provided in the literature [33,34].

No melt/fluid inclusions were observed in beryl I. A large number of melt, fluid, and solid mineral inclusions were found in beryl IIa and IIb. To more accurately define the fluid properties during the crystallization of beryl IIa and IIb, the fluid inclusions hosted in the shallow gray quartz that coexisted with the beryl IIa and IIb were also selected for microthermometric measurement. Two types of melt inclusions and four types of fluid inclusions were identified in beryl IIa, IIb, and quartz.

Melt inclusions can be divided into water-bearing melt inclusions (Type A) (Figures 7a and 8a) and water-poor melt inclusions (Type B) (Figure 7b).

Four types of fluid inclusions were recognized. Type 1: gas-liquid two-phase inclusions (Figure 7d). Type 1 inclusions mainly hosted by quartz are mostly 10–30 μm in size, and the proportion of gas is generally less than 20%, some up to 40%. This kind of inclusion can be divided into primary and secondary inclusions. The primary inclusions occur as isolated inclusion or grouped in the three-dimensional arrays (Supplementary material S2 Figure S1). Some type 1 inclusions coexist with $CO_2$ three-phase inclusions.

Type 2: Aqueous–carbonic three-phase inclusions. Type 2 inclusions are developed in both beryl II and quartz (Figure 7c–f). The type 2 primary inclusions are mostly parallel to the c-axis of beryl II (Supplementary material S2 Figure S1). At room temperature (30 °C), they are characterized by gaseous $CO_2$, liquid $CO_2$, and brine. $CO_2$-NaCl-$H_2O$ three-phase inclusions in quartz are round or oval, and most of them are in the range of 10–30 μm, commonly the necking down phenomenon (Figure 7e).

Type 3: Pure liquid-phase $CO_2$ inclusions. It is commonly found in beryl, with a size around 20 μm (Figure 7c,d). At room temperature (30 °C), it only presents the liquid $CO_2$ phase, and $CO_2$ bubbles appear when the temperature drops below −5 °C.

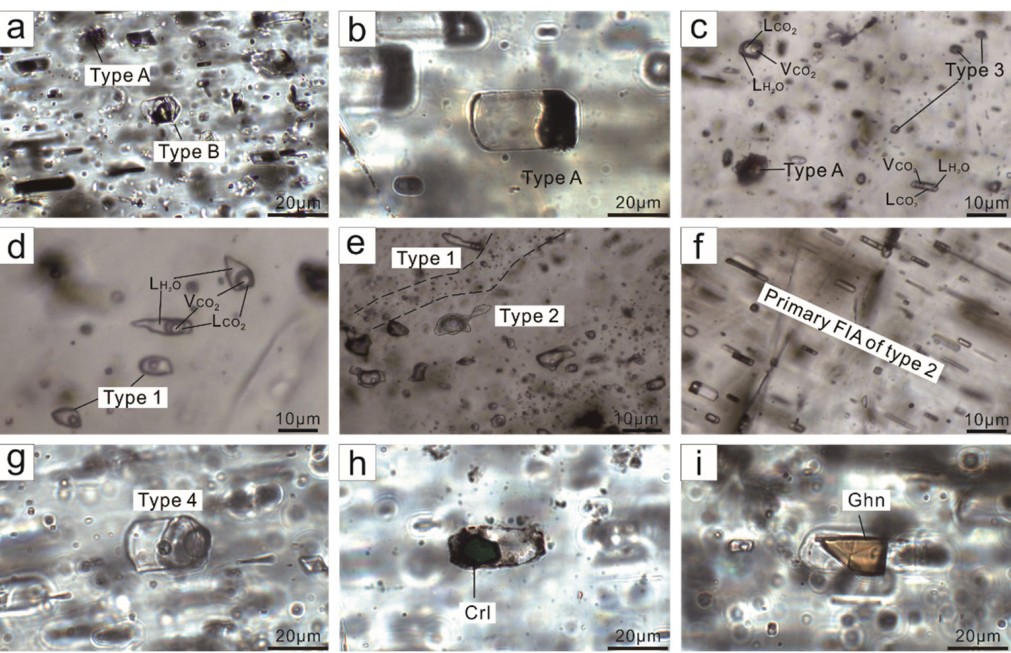

**Figure 7.** Transmitted-light photomicrographs of melt inclusions (MI), fluid inclusions (FI), and mineral inclusions in beryl IIa, IIb, and quartz crystals. (**a**) Type A and B inclusions in beryl IIa (20DK1-4); (**b**) Type B water-poor MI in beryl II (20DK1-4); (**c**) Three different inclusions in beryl IIa (type A, type 2, and type 3) (20DK1-4); (**d**) two different inclusions in quartz (type 1 and type 3); (**e**) two different inclusions in quartz (type 1 and type 2), part of type 2 FI appear to the necking-down phenomenon; (**f**) Primary type 2 fluid inclusions assemblage (FIA) are parallel to c-axis of beryl IIb(20DKLS1-1); (**g**) Silicate daughter mineral-bearing inclusion (type 4 FI) hosted by beryl IIa (20DK1-4); (**h**) Solid cryolite (Crl) inclusion in beryl IIa (20DK1-4); (**i**) Solid gahnite (Ghn) inclusion in beryl IIa (20DK1-3).

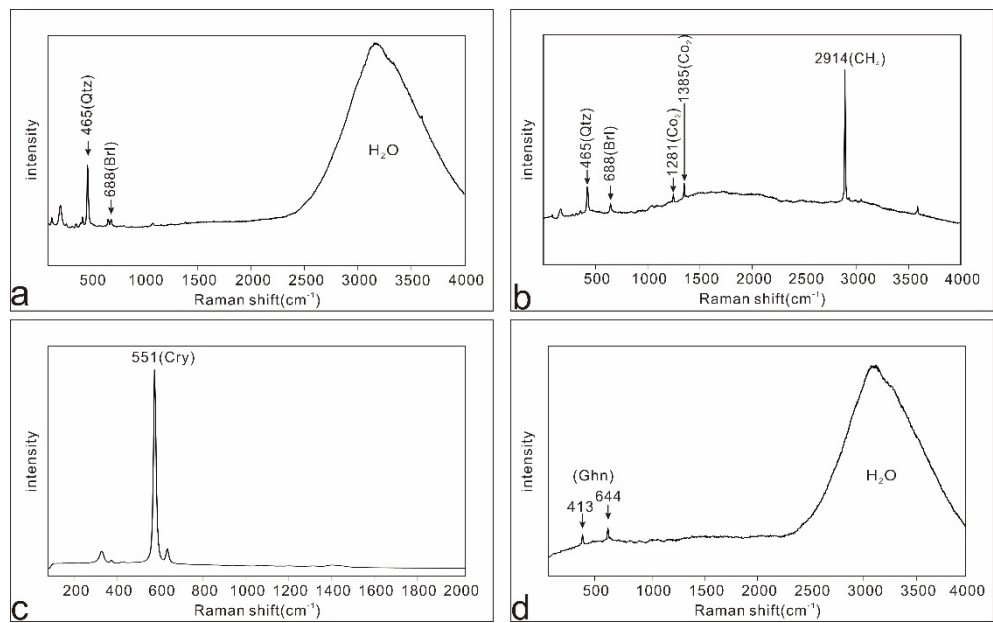

**Figure 8.** Raman spectra of (**a**) Solid phase, probably quartz and rich-aqueous in a melt inclusion form beryl II; (**b**) The vapor phase forms typical primary fluid inclusion hosted by beryl II contains $CO_2$ and $CH_4$; (**c**) Cryolite solid inclusion hosted by beryl II and quartz; (**d**) Gahnite solid inclusion hosted by beryl II.

Type 4: Silicate daughter mineral-bearing inclusions (Supplementary material S2 Figure S2), at room temperature, there are four phases: solid silicate daughter minerals, gaseous $CO_2$, liquid $CO_2$, and brine solution (Figure 7g). Silicate daughter minerals account for less than 20% of the volume of inclusions. Surrounded by the fluid phase, it is located at the edge of the inclusion, and the proportion of $CO_2$ is 20%~50%.

The solid inclusions are identified as cryolite and gahnite (Figure 7h,i) on the basis of their Raman spectra (Figure 8c,d).

Before measurements, fluid inclusion populations were classified and described using a petrographic microscope and Raman spectrometer analyses. Microthermometric freezing and heating measurements were performed on a broken chip of 100- or 300-μm-thick, doubly polished sections using a Linkam THSMG 600 heating–cooling stage that was calibrated with synthetic fluid inclusions; measurements of ±0.1 °C at −56.6 °C and 0 °C, and ±1 °C at 374 °C were uncertain. The phases encountered during freezing and heating experiments (from −195 to +550 °C) were solid $CO_2$, ice, hydro-halite ($NaCl \cdot 2H_2O$), clathrate ($CO_2 \cdot 4.75H_2O$), halite, other daughter minerals or incidentally trapped solid inclusions, carbonic liquid and vapor, and aqueous liquid and vapor. The phase assemblage varied as a function of the temperature of observation, composition, and density of each inclusion. The volume fractions of the coexisting phases were visually estimated at appropriate temperatures. For example, the proportion of liquid $CO_2$ to vapor $CO_2$ (degree of fill) was estimated at the temperature of melting of solid $CO_2$ (~−57 °C), or the ratio between a $CO_2$-rich phase and an $H_2O$-rich phase was estimated at the homogenization temperature of the $CO_2$ bubble (0 to 31 °C). Uncertainties in visual estimates may be especially large for irregular inclusions or those whose long dimensions were perpendicular to the microscope stage. Therefore, such inclusions were avoided whenever possible [35].

Total homogenization temperatures were reproduced to ±2.0 °C and final ice melting temperatures, final clathrate dissolution temperatures, and $CO_2$ homogenization temperatures to ±0.3 °C. For high accuracy, we used repeated warming and cooling cycles just below the expected phase dissolution or homogenization temperature [36]. The final clathrate–dissolution temperature was used to estimate the salinity of inclusions only when the disappearance of clathrate occurred in the presence of both liquid and vapor $CO_2$ [36]. Inclusions with no reproducible original vapor/liquid ratio and/or homogenization temperatures were ignored due to suspected leakage. The point of final homogenization to vapor was difficult to observe for inclusions with a very large volumetric $CO_2/H_2O$ ratio because the gradually disappearing liquid wet the walls. Therefore, measurements were made on inclusions with narrow reentrants or angular shapes [37]. Detailed phase transition between melt-inclusion and fluid inclusion upon heating is summarized in the Supplementary material S2 (Figures S3–S8).

The FLUIDS computer package was used to calculate fluid properties [38,39]. Calculations were performed in the $NaCl$-$H_2O$ and $CO_2$-$NaCl$-$H_2O$ fluid systems, depending on the estimated fluid inclusion composition.

## 4. Results

### 4.1. Beryl Chemical Compositions

The major and trace element data for the Dakalasu No.1 pegmatite beryl (Supplementary material S1) show minor variations between beryl I, beryl IIa and beryl IIb. The contents of $SiO_2$, $Al_2O_3$, BeO (calculated), $Na_2O$, $Cs_2O$, MgO, and FeO of beryl I are in the range of 66.25–68.182 wt%, 16.22–16.93 wt%, 12.25–12.76 wt%, 1.653–2.096 wt%, 0.006–0.066 wt%, 0.058–0.268 wt%, and 0.52–0.981 wt%, respectively. The beryl IIa has lower contents of $SiO_2$ (61.53–66.72 wt%), $Na_2O$ (0.38–0.93 wt%), MgO (bdl–0.22 wt%), and FeO (0.3–0.79 wt%), but higher BeO (12.07–13.43 wt%) and $Cs_2O$ (0.03–0.23 wt%) contents than those of beryl I. The contents of $SiO_2$, $Al_2O_3$, BeO, $Na_2O$, $Cs_2O$, and FeO of beryl IIb are similar to those of beryl IIa and are in the range of 63–66.41 wt%, 16.04–16.97 wt%, 12.40–13.46 wt%, 0.33–0.85 wt%, 0.06–0.22 wt% and 0.35–0.63 wt%, respectively. Beryl I, IIa, and IIb show variations of Li (304.98–688.23 ppm), Cs (890.21–2139.39 ppm), Mg

(123.49–1806.64 ppm), Mn (22.74–271.21 ppm), Ti (71.98–158.52 ppm), K (31.74–745.01 ppm), Sc (bdl–13.74 ppm), V (0.45–7.95 ppm), Rb (18.37–202.46 ppm) and Ga (17.1–74.12 ppm). The Li and Cs contents increase from the beryl I, beryl IIa to beryl IIb, while Mg, K, and Rb contents decreased. Beryl IIa is enriched in Ti. The concentrations of B, Co, Sr, Zr, Nb, Mo, Sn, Ba, Hf, Ta, Pb, and Bi were always below the limit of detection, and are, therefore, not reported here.

### 4.2. Fluid Inclusion Microthermometry

The microthermometry of type1, 2, and 3 inclusions was carried out. Type 4 inclusions have a very difficult to determine homogenization temperature because they are easy to crack during heating. Microthermometric characteristics are summarized in Table 3 and shown in Figure 9.

**Table 3.** Summary of inclusion types from the Dakalasu No.1 Pegmatite.

| Host Mineral | Inclusion Type | Genetic Type | $T_{m,ice}$ (°C) | | $T_{m,clath}$ (°C) | | Salinity (wt% NaCl equ.) | | $T_{h,tot}$ (°C) | | State | n |
|---|---|---|---|---|---|---|---|---|---|---|---|---|
| | | | Range | Mdn ± σ | Range | Mdn ± σ | Range | Mdn ± σ | Range | Mdn ± σ | | |
| Beryl II | Type 1 | Primary | −4.1 to −2 | −3.1 ± 1.1 | | | 3.4–6.2 | 5.1 ± 1.5 | 300–320 | 310 ± 7 | L | 3 |
| | Type 2 | Primary | | | 5–9.2 | 6.1 ± 0.9 | 1.6–9 | 7.2 ± 1.7 | 300–400 | 340 ± 35 | L | 53 |
| | Type 3 | Pseudosecondary | | | 5.2–6.2 | 5.6 ± 0.6 | 7.1–8.7 | 7.9 ± 0.4 | 260–275 | 268 ± 7 | L | 7 |
| Quartz | Type 1 | Primary | −5.5 to −5 | −5.2 ± 0.2 | | | 7.8–8.6 | 8.1 ± 0.3 | 340–350 | 345 ± 5 | L | 5 |
| | Type 2 | Primary | | | 5.2–7 | 6.1 ± 0.7 | 5.7–8.7 | 7.1 ± 1.5 | 315–385 | 342 ± 35 | L | 19 |
| | Type 3 | Primary | | | 7.2–7.3 | 7.2 ± 0.1 | 5.1–5.3 | 5.2 ± 0.1 | 280–290 | 285 ± 5 | L | 2 |

Notes: $T_{m,ice}$ = final ice melting temperature, $T_{m,clath}$ = final clathrate melting temperature, $T_{h,tot}$ = homogenization temperature. L = liquid. Mdn = median, σ = standard deviation, n = total number of the microthermometric measurements.

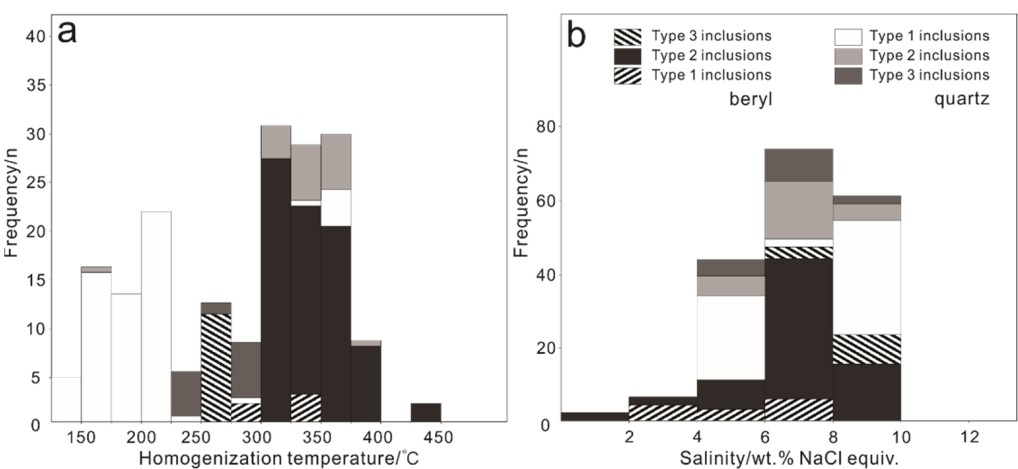

**Figure 9.** Frequency distribution of (**a**) total homogenization temperature; (**b**) salinity.

The final ice melting temperature of primary type 1 inclusions in the range of from −4.1 to −2 °C and −5.5 to −5 °C is observed in beryl II and quartz, indicating a low salinity of 3.4–6.2 wt% NaCl equivalent and 7.8–8.6 wt% NaCl equivalent [40]. Total homogenization temperatures of primary type 1 inclusions are observed in beryl II (to liquid, 310 ± 7 °C) and in quartz (to liquid 345 ± 5 °C).

The final $CO_2$ melting temperatures of primary type 2 inclusions are observed in beryl II in the range between −65.8 °C and −57.3 °C, in quartz in range between −66.3 °C and −56.4 °C. The final clathrate melting temperature of $CO_2$ ranges from 5 to 9.2 °C in beryl II and 5.7 to 7 °C in quartz, suggesting a salinity between 1.6 and 9 wt% NaCl equivalent in beryl II, 5.7 and 8.7 wt% NaCl equivalent in quartz [41]. The homogenization of $CO_2$ (to liquid $CO_2$) in the type 2 inclusions was observed in beryl II in the range between 20.2 °C and 27.3 °C, in quartz in the range between 15 °C and 28.4 °C. The total homogenization

temperatures of primary type 2 fluid inclusions are observed in beryl II in the range between 300 °C and 400 °C and in quartz in the range between 315 °C and 385 °C.

The type-3 inclusions have a liquid-vapor two-phase at room temperature but become carbon-rich, three-phase inclusions when frozen to −5 °C. The final $CO_2$ melting temperatures of type 3 inclusions are between −62.3 °C and −61.2 °C in beryl II, the temperature of the inclusions is observed in quartz in the range between −64 °C and −63 °C. Final clathrate melting temperatures range from 5.1 to 7.3 °C suggest a salinity between 5.1 and 8.7 wt% NaCl equivalent [41]. The homogenization of $CO_2$ (to liquid $CO_2$) occurs in the range between 17.5 °C and 27.3 °C. Homogenization temperatures of type 3 inclusions in the range of 260 to 290 °C are observed in beryl II and quartz.

## 5. Discussion

### 5.1. Beryl as Potential Indicators of Pegmatite Evolution

The chemical composition and internal structures of beryl may be used to indicate formation and evolution degree of pegmatite [42–44].

Variations in beryl composition, especially in alkaline and ferromagnesian element concentrations, could be indicative of mineral and host-rock fractionation [45]. The fractionation trend of beryl, toward Na, Li, Cs-rich, and Fe-Mg-poor composition, is well-documented in rare-element pegmatites worldwide [7,8,10,46]. However, this general trend could be more complex in some pegmatites. In recent years, The Na/Cs and Na/Li ratios of beryl are used to indicate the degrees of pegmatite evolution [7,8,42].

The Na/Cs ratio and FeO content of different types of beryl (I, IIa, and IIb) from the Dakalasu No.1 pegmatite reveal that the averages of Na/Cs ratio (79.61) and FeO content (0.057%) for beryl I are higher than those for beryl IIa (Na/Cs = 4.05; FeO = 0.037%) and beryl IIb (Na/Cs = 3.44; FeO = 0.041%). The differences in Na/Cs ratio and FeO content in beryl I, IIa, and IIb possibly indicate that the three types of beryl were formed in different stages of the evolution of pegmatitic melt.

Beryl I occur in the wall zone, beryl IIa, and IIb occur in the intermediate zone of the Dakalasu No.1 pegmatite. Na/Cs ratio versus FeO content of beryl indicate that the evolution degree of the wall zone in the Dakalasu No.1 pegmatite is similar to the zone of I, II, and IV of the Koktokay No.3 pegmatite. The intermediate zone of Dakalasu No.1 pegmatite is similar to beryllium mineralized pegmatite in the Nanyang Mountain mining area (Nanyangshan No.302, Wayaogou and Xishangou). The evolution degree of Dakalasu No.1 pegmatite is lower than that of lithium mineralized pegmatite (Nanyangshan No.703, Daxigou and Jiucaigou) and the zone IV of the Koktokay No.3 pegmatite (Figure 10). In addition, the FeO content of beryl in Dakalasu No.1 pegmatite has a linear positive correlation with Na/Cs ratio, which is similar to the beryl in Koktokay No.3 pegmatite [7]. This may be due to the crystallization of iron-containing mineral phases (columbite-group minerals, tourmaline, and muscovite, etc.) in the system, which consumes the iron in magma and reduces the FeO content in residual magma, and the subsequent crystallization of beryl IIa (0.037%) and IIb (0.041%) showing a lower FeO content than beryl I (0.057%). Therefore, the FeO content of beryl can be used as a potential index of magma differentiation and evolution.

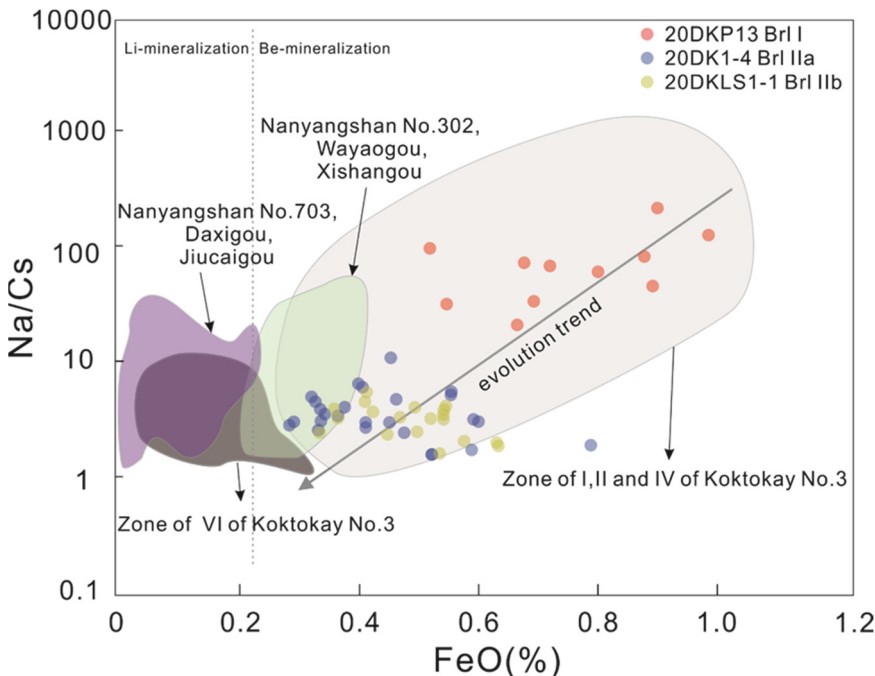

**Figure 10.** Plots of FeO vs. Na /Cs. The contents of FeO and Na /Cs values of beryl from the Koktokay No.3 pegmatite and the East Qinling Lushi pegmatites are from [7,8].

Lithium and Cs contents increase from the beryl I, beryl IIa to beryl IIb, while Mg and Rb contents decreased. Scandium, V and Ga contents greatly differed in beryl I and beryl II, while these elements are similar in beryl IIa and IIb. Titanium is enriched in beryl IIa (Figure 11). The variation in Li, Cs, and Mg contents in beryl I, IIa, and IIb are closely related to the crystallization evolution of pegmatite residual melts [4]. The change in Mg contents in beryl IIb compared with beryl IIa may be due to the metasomatism of beryl IIa under the influence of later exsolving fluids or external fluids, which makes Mg migrate to other minerals, e.g., muscovite in beryl IIb fracture, or nearby altered minerals (Figure 6d). The reason for Ti enrichment in beryl IIa may be related to the F content in the melt; when beryl IIa was formed, fluorine was in a saturated state, as evidenced by the presence of cryolite ($Na_3AlF_6$) in beryl IIa (Figure 7h). Fluorine-rich fluids can activate and migrate Ti to the greatest extent in the magmatic-hydrothermal fluid system [47]. A high number of rutile dissolution experiments have proved that Ti migration is related to F content. The solubility of rutile in pure water is very limited (<100 μg/mL) [48,49]. Even in the presence of NaCl and/or silicate, the maximum solubility of rutile is less than 5000 μg/mL [48–50], but the solubility of rutile in F-rich fluid can reach 45739μg/mL [47,51]. The mechanism that makes rutile easy to dissolve in F-rich fluid might be Ti forming a soluble complex with F. Experiments reveal that $Ti^{4+}$ will be complexed with $OH^-$ to form $[TiO]^{2+}$, $[TiO(OH)]^+$, $[Ti(OH)_4]^0$, $[Ti(OH)_5]^-$, $[Ti(OH)_6]^{2-}$ in $H_2O$-saturated hydrothermal fluid [49,52]. When $F^-$ is contained in $H_2O$-saturated hydrothermal fluid, $F^-$ will replace part of $OH^-$ to form hydroxide-fluoride complexes of Ti, e.g., $[Ti(OH)_4F]^-$, $[Ti(OH)_3F]^0$, $[Ti(OH)_2F_2]^0$ [47]; with the increase in F concentration, $F^-$ completely replaces $OH^-$ to form $[TiF_5]^-$, $[TiF_6]^{2-}$ complexations [53].

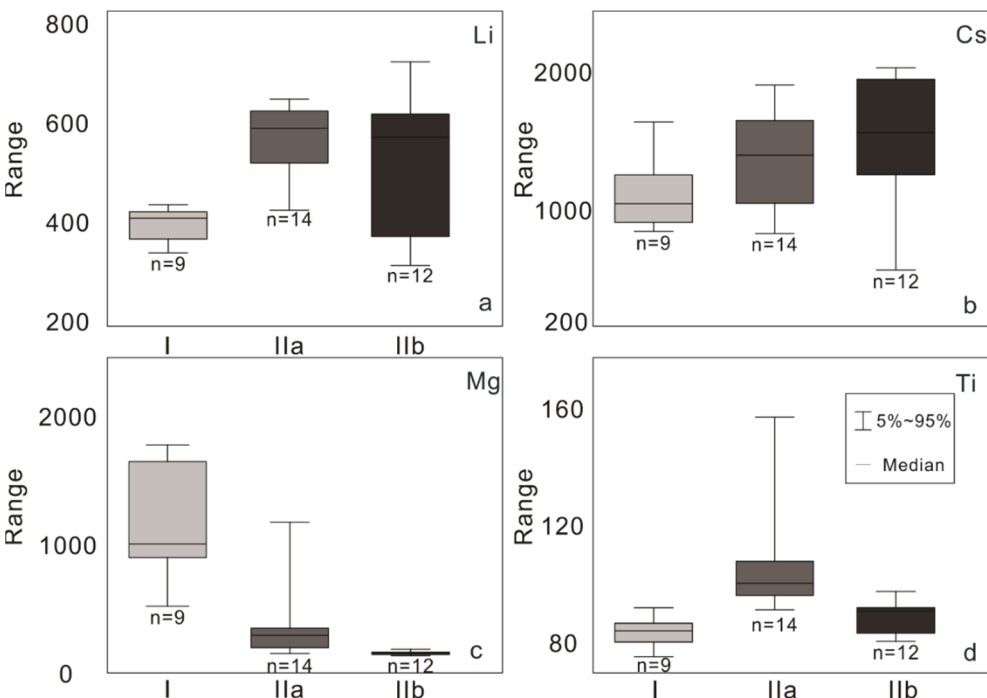

**Figure 11.** Box plots illustrating variations in Li (**a**), Cs (**b**), Mg (**c**), and Ti (**d**) concentrations of beryl (I, IIa, and IIb) from the Dakalasu No.1 pegmatite.

### 5.2. Fluid Properties

Silicate daughter mineral-bearing inclusions are not only found in the Dakalasu No.1 pegmatite, but also in Koktokay No.3 pemgatite [54,55], Tanco pegmatite deposit [56], and South China granite [54]. Silicate daughter mineral-bearing inclusions are melt-fluid inclusion in the magmatic-hydrothermal transition stage. The type inclusions in the Dakalasu No.1 pegmatite have fewer, single-type silicate daughter minerals, which may be caused by its low degree of evolution.

The liquid phase component ($CO_2$-NaCl-$H_2O$) in silicate daughter mineral-bearing inclusions and the associated $CO_2$-NaCl-$H_2O$ inclusions have similar gas–liquid ratios, solid $CO_2$ melting temperature, homogenization temperature, and salinity. It can be inferred that $CO_2$-NaCl-$H_2O$ is separated from the melt represented by silicate daughter mineral-bearing inclusions [57]. On the other hand, melt inclusions and silicate daughter mineral-bearing inclusions are found in the beryl IIa and IIb, suggesting that the silicate daughter mineral-bearing inclusions are the result of fluid exsolution from the melt.

The $CO_2$ clathrate melting temperatures $Tm_{calth}$ of the $CO_2$-NaCl-$H_2O$ (type 2) in beryl and quartz were recorded within between 5 and 9.2 °C. The homogenization of $CO_2$ (to liquid $CO_2$) occurs in the range between 15 and 28.4 °C. The $CO_2$ triple point temperature $Tm_{CO_2}$ ranges between $-66.3$ °C and $-57.3$ °C, on the basis of the $_(Tm_{CO_2})$ data and estimates of the ratio (degree of fill) between a $CO_2$-rich phase and a $H_2O$-rcih phase at the homogenization of $CO_2$ (to liquid $CO_2$, 15 to 28.4 °C), according to the formula [58]. $X_{H_2O}$, $X_{CO_2}$, and $X_{NaCl}$ were calculated, and then projected into the $CO_2$-$H_2O$-10 × NaCl ternary diagram to estimate the captured pressure and temperature of inclusions [35,59]. The minimum trapped pressure (temperature) of $CO_2$-NaCl-$H_2O$ inclusions of beryl and quartz is 200 Mpa (500 °C) (Figure 12). Under this condition, primary beryl and quartz are crystallized from the melt, and then fluid phases exsolve from the melt.

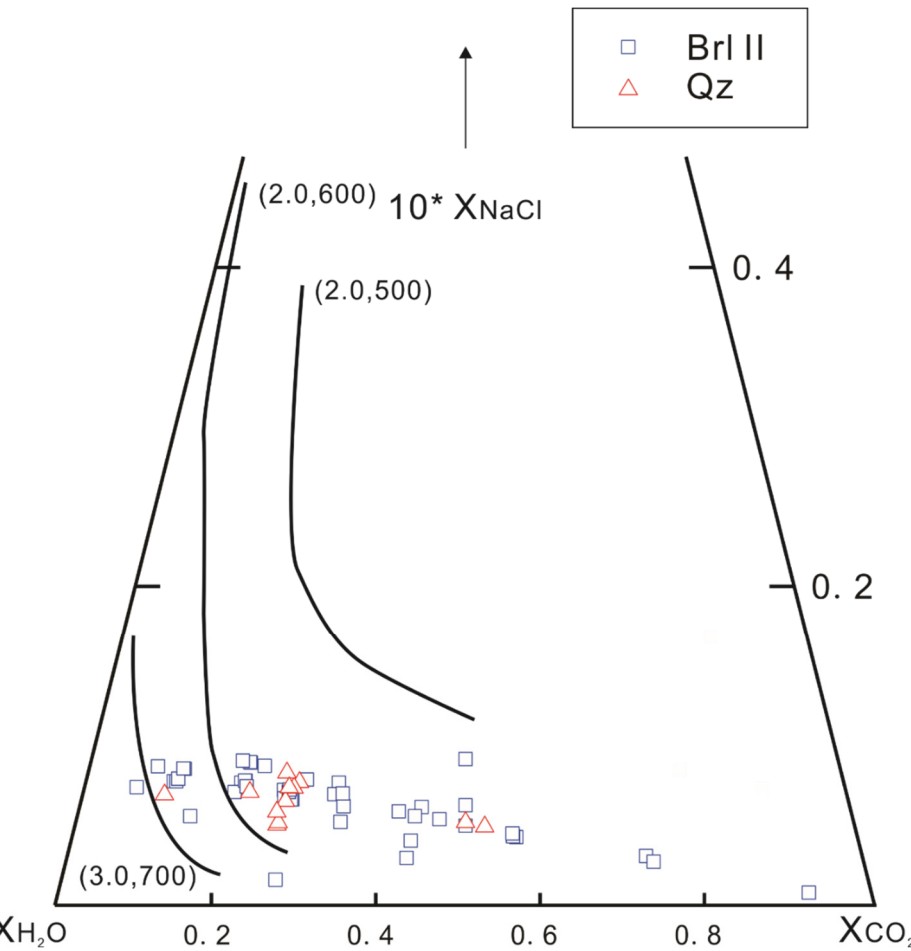

**Figure 12.** $CO_2$-$H_2O$-10×NaCl ternary (mole fraction) with Dakalasu No.1 pegmatite (beryl II and quartz) type 2 inclusions trapped in the one-fluid phase region, shown by the shaded area. Note that the diagram is stretched along the vertical axis (after [35]). Thick curves represent isothermal-isobaric projections of the solvi bracketed in experiments of Frantz et al. [59], shown as half-filled squares. P in kbar and T in °C for the solvus lines and for selected experiments are shown in parentheses.

The $CO_2$-NaCl-$H_2O$ hydrothermal fluid was exsolved in the pegmatitic melt. However, due to the low mutual solubility between $CO_2$ and $H_2O$, $CO_2$ content gradually reached saturation with the decrease in temperature and pressure, which led to a $CO_2$-rich fluid and NaCl-$H_2O$ being separated from the original $CO_2$-NaCl-$H_2O$ fluid. Previous studies have found that when $CO_2$-NaCl-$H_2O$ is separated, $CO_2$-rich fluid inclusions with low salinity and brine inclusions with high salinity will coexist [60–62]. However, no high-salinity inclusions were detected in the gas–liquid, two-phase inclusions measured by the Dakalasu No.1; the salinity range of these inclusions is from 7.86 to 8.55 wt% NaCl equivalent. The main reason for this may be the low evolution degree of pegmatite, as pegmatite melts are rich in F. Increasing F content of the melt reduces the partitioning of Cl into the fluid, so that later exsolving fluids have low salinity (at unchanged pressure conditions) [63].

*5.3. Enrichment Mechanism of Beryllium*

The beryllium content of chondrite is considered to represent the bulk composition of the earth, which is less than 0.1 ppm, and the beryllium content in the crust is only 0.001% [64]. There are two different mechanisms for the strong enrichment of Be: (1) melt–melt immiscibility (2) extreme level of crystallization differentiation [11,65,66]. As far as the field observation and mineralization type of the Dakalasu No.1 pegmatite (Be-Nb-Ta) is concerned, its evolution degree is not at the extreme level. The observed inclusion

assemblage in beryl IIa and IIb is more consistent with the first mechanism: melt–melt immiscibility (Figure 7a,b).

In addition, to Be enrichment caused by melt-melt immiscibility, the intrusion mode and evolution duration of the Dakalasu No.1 dike also affect the enrichment of Be. The Dakalasu No.1 pegmatite intrudes in the bedding plane and joints where granite develops. When granite bedding occurs, it should be in the shallow part of the crust [67]. From the perspective of mechanical properties, the appearance of bedding requires an elastic response of rock [68]. This means that the top of the Dakalasu porphyritic biotite granite is rigid when bedding occurs. Therefore, the porphyritic biotite granite may have been in a cooling state when the granitic melt that forms the Dalakasu No.1 pegmatite intrudes into the bedding [69]. The Dakalasu porphyritic biotite was dated at $261.4 \pm 2.1$ Ma ($2\sigma$) using the zircon U–Pb method [28], which is older than the age of the wall zone of Dakalasu No.1 pegmatite. Wang et al. [14] obtained a muscovite $^{40}Ar/^{39}Ar$ plateau age of $248.4 \pm 2.1$ Ma and Zhou et al. [18] reported a columbite U–Pb age of $239.6 \pm 3.8$ Ma for the wall zone of Dakalasu No.1. Therefore, the age difference between granite and pegmatite also supports the assumption that the surrounding rock temperature of granite was lower when pegmatite was emplaced. When the melt of the Dakalasu No.1 pegmatite is emplaced, the temperature of porphyritic biotite granite as surrounding rock should be much lower than that of pegmatite melt after a cooling time of more than 10 Ma. According to the cooling model [46], the huge temperature difference at this time leads to the rapid cooling of pegmatite melt, which leads to the rapid crystallization of beryl I under solidus. The intermediate zone of No.1 pegmatite was the date from $229.0 \pm 1.0$ Ma to $228.4 \pm 0.3$ Ma using the columbite U–Pb method [16,18]. The age data indicate that there is a distinct gap (20~11 Ma) between the crystallization ages of the wall zone and the intermediate zone of No.1 pegmatite. The partition coefficient of beryllium in most rock-forming minerals is less than 1 (quartz $C_B^{mineral(min)}/C_{Be}^{melt} = 0.24$, K-feldspar $C_{Be}^{min}/C_{Be}^{melt} = 0.3$, plagioclase: An1-5 $C_{Be}^{min}/C_{Be}^{melt} = 0.10$ and An10-15: $C_{Be}^{min}/C_{Be}^{melt} = 0.6$) [70]. The intermediate zone formed after a long time of residual pegmatite melt evolution, beryllium is further enriched in the residual melt to form large and euhedral beryl crystals in the intermediate zone.

## 6. Conclusions

(1) The occurrence position of beryl in the Dakalasu No.1 pegmatite, mineral paragenetic association, and trace elements of beryl indicate that there are at least two periods of beryl mineralization in the Dakalasu No.1 pegmatite.

(2) The crystallization of melt at highly undercooled states and melt–melt–fluid immiscibility are inferred to be the cause of Be enrichment and mineralization in the Dakalasu No.1 pegmatite.

**Supplementary Materials:** The following supporting information can be downloaded at: https://www.mdpi.com/article/10.3390/min12040450/s1, Table S1 in Electronic Supplementary Material S1 (ESM 1) and Figures S1–S8 in Electronic Supplementary Material S2 (ESM 2).

**Author Contributions:** Formal analysis, Q.S.; investigation, Q.S., Y.L., C.L., H.F., C.C., Y.B., P.S. and H.P.; resources, P.S.; data curation, Q.S.; writing—original draft preparation, Q.S.; writing—review and editing, C.L., H.F.; visualization, Y.L.; supervision, P.S. All authors have read and agreed to the published version of the manuscript.

**Funding:** This work was granted by the National Natural Science Foundation of China (91962213), Regional Collaborative Innovation Project, Xinjiang (2020E01043), the International Partnership Program of International Cooperation Bureau, Chinese Academy of Sciences (132A11KYSB20190070), and the National Key R&D Program of China: (2018YFC0604004).

**Data Availability Statement:** The data presented in this study are available in this article.

**Acknowledgments:** We are grateful for the valuable support from the Natural Resources Bureau of Altai Prefecture for access to samples and information about the Dakalasu No.1 pegmatite. We are

grateful to Lihui Jia, Shitou Wu, Xiaoguang Li, and Yingxia Xu for their assistance in EMPA, LA-ICP-MS, SEM, and Raman spectroscopy, and Microthermometry. Finally, this manuscript benefited considerably from the comments and suggestions of the Editor and three anonymous reviewers.

**Conflicts of Interest:** The authors declare no conflict of interest.

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
