# Peer review of "Beryl Mineralogy and Fluid Inclusion Constraints on the Be Enrichment in the Dakalasu No.1 Pegmatite, Altai, NW China"

_minerals, doi:10.3390/min12040450_

Round 1
Reviewer 1 Report
Suo et al. present new Beryl chemistry and fluid inclusion data to investigate beryllium mineralization in rare metal pegmatite intrusion from the China Altai orogen, indicating petrogenesis of such type Be-Nb-Ta deposits hosted in pegmatites in the region. This study is well written and has implications for further exploring rare metals in Altai and elsewhere, thus is worthy of being published in the journal if some problems are fixed via a minor revision.
- An explanation is required what ‘melt-melt-fluid immiscibility’ is and how such process is related to beryllium mineralization.
- Have you observed the phase relationship (or transition) between melt-inclusion and fluid inclusion upon heating?
More specific are listed below. Detailed comments are marked in the annotated PDF manuscript for authors’ consideration.
Figure 1: Legend needs be rearranged in sequential order in time, putting the youngest unit on the top and oldest at bottom. The same problem is evident in Figure 2, requiring to be fixed.
Lines 137-142: Information on ‘Where is EMPA performed?’ seems necessary.
Table 2: Where is the significant decimal for the analytical data? For example, ‘399.53’ as reported in Table 2, should read as ‘399’ or ‘400’.

Author Response
We have modified the content in the PDF you provided, thank you very much.
Response to Reviewer 1 Comments
- An explanation is required what ‘melt-melt-fluid immiscibility’ is and how such process is related to beryllium mineralization.
Reply:
The observed complex inclusion population in the Dakalasu No.1 beryl is further evidence that during pegmatite forming processes at a relatively high crustal level melt-melt and melt-fluid immiscibility are real processes. Type A (relatively H2O-poor) and Type B (extremely H2O-) melt inclusions result from a melt–melt immiscibility event, which induced development of two immiscible conjugate melts with contrasting properties along the opening solvus, due to cooling and concomitant fractional crystallization from 700 to 500°C (Thomas et al. 2000).
Zajacz (2007) measured a mean of 570 ± 150 ppm Be (n = 22) with LA-ICP-MS in unheated type-A MI entrapped in pegmatitic quartz from Mt. Malosa in the Chilwa alkaline province in Malawi. There, he found a type-B MI with 1723 ppm Be. Clearly, some silicate melts are capable of dissolving high concentrations of Be. B-, F-, P-, ± Li-enriched magmas during extended fractional crystallization and late-stage melt–melt immiscibility will facilitate even greater Be enrichments in the comparatively H2O-rich melt fractions represented by type-B MI (Thomas et al. 2011).
Figure 1. Coexistence of melt and fluid inclusion in beryl II
As the residual magma of pegmatite continues to crystallized, fluid exsolve occurs, leading to the immiscibility of melt and fluid. Evidence for coexistence of melt and fluid inclusion (Fig .1). Evensen et al. (1999) provided important constraints for compositionally simple granitic melts and suggested that Be re-mobilization leading to mineralization largely involves lower-temperature secondary (fluid) processes. Thomas et al. 2011 also draw additional conclusions regarding the speciation of Be in pegmatite-forming melt systems from investigation of the Be-bearing daughter mineral phases in the most H2O-rich melt inclusions. In the case of evolved volatile and H2O-rich pegmatite systems, B, P, and carbonates are important for the enrichment and formation of stable Be complexes.
- Have you observed the phase relationship (or transition) between melt-inclusion and fluid inclusion upon heating.
Reply:
Phase transition of fluid inclusions during microthermometry has been put into new supplementary materials (ESM 2). Figure. 2~5, 7 beryl II; Figure.6 quartz.
Figure 2. phase relationship when final CO2 melting.
Figure 3. phase relationship when final CO2 clathrate-dissolution.
Figure 4. phase relationship when homogenization to liquid CO2.
Figure 5. phase relationship when total homogenization to vapor CO2.
Figure 6. phase relationship during microthermometry of inclusions in quartz
Figure 7. phase relationship of MI? in beryl II
Point 1: Figure 1: Legend needs be rearranged in sequential order in time, putting the youngest unit on the top and oldest at bottom. The same problem is evident in Figure 2, requiring to be fixed.
Response 1: Thanks, we have corrected them in the revision on Figures 1 and 2.
Point 2: Lines 137-142: Information on ‘Where is EMPA performed?’ seems necessary.
Response 2: Thanks for pointing this out. In the revision, we have added it on Line 165-167
Point 3: Table 2: Where is the significant decimal for the analytical data? For example, ‘399.53’ as reported in Table 2, should read as ‘399’ or ‘400’.
Response 3: Thanks, we have corrected them in the revision on Table 2
Reference
- Thomas, R.; Webster, J.D.; Heinrich, W. Melt inclusions in pegmatite quartz: complete miscibility between silicate melts and hydrous fluids at low pressure. Contrib Mineral Petrol. 2000, 139, 394–401.
- Zajacz Z (2007) Mass transfer during volatile exsolution in magmatic systems: insights through methodological developments in melt and fluid inclusion analysis. Unpublished Dissertation No. 17254, ETH Zu¨rich, 183 p
- Thomas, R., Webster, J.D.; Davidson, P. Be-daughter minerals in fluid and melt inclusions: implications for the enrichment of Be in granite-pegmatite systems. Contributions to Mineralogy & Petrology. 2011, 161, 483–495.
- Evensen, J.M.; London. D.; Wendlandt, R.F. Solubility and stability of beryl in granitic melts. Am Mineral 1999, 84, 733–745.

Reviewer 2 Report
Shou et al. 2022 review
Overall, this manuscript does contain some publishable results, but it needs a fair amount of work to be acceptable for MINERALS. My greatest concern has to do with the fluid inclusion data and interpretations. I think a lot more effort needs to be expended on describing the textures used to infer a primary origin as well as the inferences of fluid immiscibility or no immiscibility. I don’t think the data on the secondary fluid inclusions tells your readers anything about the formation of beryl and quartz. I would delete if from the paper.
Below are my specific comments:
Line 040: please change “which” to “that”
Line 042: please change “Lithium” to “lithium”
Line 048: weight percent?
Line 049: reference for this information?
Line 054: please delete “On the other hand,”
Line 057: please change “Ne” to “Nb” Also, most important compared to what?
Line 073: please change “discussed” to “inferred”
Line 092: what is a typical size of the pegmatite . . . width and length at surface?
Line 117: can you add a space at the beginning and end of each label in Figure 3? It would make each label much easier to read. Look at “Brl” in c and d, “Intermediate in 3b.
Line 118-123: can you state the length of the “scale’ in photos b,c,d? For example, how long is the hammer in 3b and 3d? The pen length in 3c?
Line 122: is it only beryl and tourmaline in 3d? I am guessing the tourmaline is black in 3d, but is the white mineral microcline and the gray mineral quartz? It’s hard to tell. I would describe 3d different than “Beryl coexists with a great deal of tourmaline.” Maybe “Beryl coexisting with abundant tourmaline (black), microcline (white) and quartz (gray).”
Line 124: Figure 4 captions needs: 1) what does CGM stand for? 2) granulargranite is two words. 3) why the two symbols for quartz? 4) why the two symbols for microcline? What do each indicate? Do the different size symbols refer to crystal grain size. Please inform the reader.
Line 128-130: so this is the first time you are describing beryl I, beryl IIa and beryl IIb. How are each of the three beryl types defined? What makes them different?
Line 137: where was the EMPA work done?
Line 144: at the top of table 1, what are the numbers in parentheses after the sample number? If they represent the number of analyses, then please provide some statistics on the variation of the data (maybe 1 sigma or 2 sigma error bars on each oxide). Same thing regarding table 2.
Line 147: please change “form” to “from”
Line 166: resolution of what?
Line 170: please change “Mirothermometric” to “Microthermometric”
Line 174-179: Instead of describing the phases encountered while heating and cooling, ca you just tell the readers what phases are present at room temperature? And what kind of variation do you see at room temperature? I see where you describe the inclusion types later on, but it belongs up front.
Line 180-183: are the estimated phase ratios used later on?
Line 186-187: please refer to them as “final Ice melting temperatures” and “final clathrate dissolution temperatures.” Recall that both ice and clathrate are melting until they disappear.
Line 193-195: please describe the homogenization behavior of all the inclusion fluids. Homogenization to liquid? Homogenization to vapor? It sounds like the latter.
Line 200-213: To the reader, these are not Results . . . instead these are descriptions of the beryl in hand specimen. They belong up front.
Line 215-219: Can you please take better photos of Fig. 5c and 5d. Nearly half of each is covered in a shadow!
Line 252-253: May I ask how do you know these are melt inclusions? The Type B inclusions could be multiple-solid inclusions that were not trapped as melt? What evidence to you have that these are melt inclusions?
Line 255: please change “inclusion” to “inclusions” also “is” to “are”
Line 256: please change “inclusions” to “inclusion”
Line 257-260: This part needs help. You must describe the criteria for inferring primary and secondary inclusions here. Negative crystal shape versus irregular shape isn’t good enough. More information about how the inclusions occur in the quartz and beryl is needed. You must convince the reader that you are investigating inclusions that are directly associated with the formation of beryl and quartz. See Roedder’s 1984 criteria. Not sure if studying the secondary inclusions is even necessary for this paper. See below on Table 3 and Figure 9.
This is applicable for line 261 too!
Line 270-274: do you know, based on Raman spectra, what the daughter silicates are?
Line 254-274: you describe the 4 types of fluid inclusions, but you don’t write anything here about whether they constitute a fluid inclusion array (FIA) or fluid inclusion assemblage (FIA). This is important because if types 1,2, and 3 are found in the same FIA, then one could argue for fluid immiscibility. But, if they are not part of the same array, they could have been trapped at different times. This is a significant detail !!!!
Line 279: You state type A, but types A and B are shown in “a.” Also, delete the first A, you just need to write “(a) Types A and B inclusions . . . .”
Line 280: You state type B, but type A is shown in “b.” Again, delete the first A, you just need to write “(b) Type B water poor . . . .”
Line 281: a type A melt inclusion is show in image (c) rather than Type B as stated.
Line 281: are these two inclusion types part of the same array? Is it possible the two type 1 inclusions have some CO2 and you just can’t see the two bubbles because they are not flat like the two inclusions above?
Line 282: it seems that the Type 2 inclusions in image E are much more CO2 rich than those in image d. It appears that necking occurred after phase separation to yield the CO2-rich portions. The Type 1 inclusions in image E between the two dashed lines are very small, so how can you determine that no CO2 is present?
Line 283: In image F, how do you know the crystallographically controlled inclusions are fluid inclusions? Hard to tell just from images. Do you observe both the liquid and vapor CO2? I am guessing that the inclusions are parallel to the c-axis of beryl, if so, you should state this. A close-up image of one of these inclusions would help significantly.
Line 284: what is the identification of mineral or minerals in figure G? See below:
Line 288-289: How do you know this is glass? Glass has a very specific Raman spectrum and I don’t see any evidence for this. Also, you state the inclusion in 7a and 8a is from a beryl host, yet your Raman spectrum shows a quartz peak. Is this a quartz grain in the melt inclusion? Is it really coming from a quartz host instead? There are no background beryl peaks? None of this makes sense, please clarify this! I would make the same comment about the quartz peak in 8b. That is a pretty significant CH4 peak in the melt inclusion vapor!
Line 293-297: What do you mean by partial homogenization temperature? I’ve never heard such a term. Do you mean decrepitation temperature? Is this the temperature at which the inclusion exploded and leaked its constituents? Some have had success heating these inclusions in the diamond anvil cell (Schmidt et al 1999, Darling and Bassett, 2002).
Line 299: Table 3 lists about 370°C not 350°C.
Line 300: The reference for determining salinity from final ice melting temperatures is not [38]. The Darling (1991) reference is for final clathrate melting temperatures in the presence of both CO2 liq and CO2 vapor. I would use Bodnar (1993) for final ice melting temperatures.
Line 304: where is “275°C and 450°C, with a distinctive peak around 325 to 350°C” listed in Table 3?
Line 305: Are these really “final CO2 melting temperatures,” maybe, if they are that low. If close to -56.6°C, they are CO2 triple point temperatures, and if they are between -66.3°C and -57.3°C, then there is a significant quantity of something other than CO2 in the vapor bubble. Maybe it’s CH4.
Line 305-307: See below, but these are “final clathrate melting temperatures.” And the reference here is [38].
Line 307-308: WHAT? All of them show CO2 homogenization by critical behavior? None show homogenization to liquid? None show homogenization to vapor? Wow, that is remarkable. To what do you attribute this very unusual behavior? Also, 15°C and 28.4°C is NOT a narrow range! If they are pure CO2, they will homogenize by critical behavior at +31.1 C only if they have the critical density. Less if there is some CH4 or another gas in there. Is this behavior really observed?
Lines 309-314: List the temperatures of phase transformations from lowest to highest, ending with total homogenization temperatures. Also, describe the homogenization behavior . . . are the inclusions homogenizing by vapor bubble disappearance or by vapor bubble expansion? Same thing with the CO2 homogenization, vapor bubble disappearance or by vapor bubble expansion? The CO2 triple point temperatures indicate the presence of a gas other than CO2, please state this! How does this affect salinity estimates? Put the [38] reference at the end of line 313, if that is where the calculation comes from.
Table 3: Why are the fluid inclusion TYPES not listed in Table 3?
Table 3 abbreviations: Tm, ice is the “final” ice melting temperature, and Tm, clath is the “final” clathrate melting temperature. Both are melting along their respective liquidii until they disappear completely.
Table 3 – I would place the negative sign in front of the actual temperatures of final ice melting rather than above in the heading. Very confusing.
Table 3 and Figure 9: Why are you studying secondary fluid inclusions? Why not just focus on what the primary inclusions are telling you about the formation of quartz and beryl? If I were to author this paper, I would focus on providing good descriptions of why the inclusions are primary in origin and then only write about the data from the primary inclusions. What are the secondary inclusions telling us? . . . about a fluid that was trapped after the quartz and beryl formed? How useful is that to us?
Line 352: “In terms of trace elements.” is not a sentence.
Line 376: Figure 10: You plot all of the Na, Cs and Fe data from Table 1, but they are not listed in Table 1, just the averages are. If this data are in the supplemental material then please state it here.
Line 382-384: This is unclear. These pegmatites don’t have silicate melt inclusions in them? Please state this more clearly.
Line 386-387: are you saying the other pegmatites have more complex array of minerals in them, indicating a higher degree of evolution? It’s not clear.
Line 391: “inferred” rather than “judged”
Line 393: “suggesting” rather than “possibly proving” We don’t prove theories in science. We find only supporting evidence for or against.
Line 398: but you don’t say how the CO2 homogenized (to liquid or vapor). This is significant if we are to estimate the bulk density of the inclusions. And, again that is not a narrow range.
Line 399: Again, these should be showing “CO2 triple point temperatures” especially if they are anywhere near -56.6°C.
Line 400: Again, the degree of fill is useful only if we know the density of the CO2 which we cannot determine because you don’t describe the homogenization behavior.
Line 419-421: Figure 12 is incompletely described. What are the curved lines? What are the two numbers in parenthesis? Are they pressure on the H2O-CO2-NaCl solvus in kilobars and temperature in centigrade?
Line 403-404: Most importantly, are you inferring the temperature and pressure of trapping from this plot? Gosh, I hope not. Are you inferring the fluids formed on the side of the H2O-CO2-NaCl solvus? If so, what fluid inclusion evidence do you have to support the trapping of immiscible liquids? If there is no evidence of fluid immiscibility then all you can infer is that the fluids formed above the location of the H2O-CO2-NaCl solvus, not on it! This plot does nothing except to show the range of fluid compositions, and to say the fluids were trapped above the H2O-CO2-NaCl solvus.
Line 405: I am guessing you mean “crystallized from the melt” not “separated from the melt”
Line 409-410: But what is your evidence that the fluid further experience unmixing? In theory, what you are stating could happen, but what evidence to you have that it did? I don’t see any evidence of fluid immiscibility in Figure 12?
Line 433-437: I have no idea what is meant by the term “granite bedding.” Please explain.
Line 463: Please replace “the reasons for” with “inferred to be the cause of”

Author Response
Please see the attachment
Point 1: Line 040: please change “which” to “that”
Response 1: Thanks, we have corrected it in the revision on Line 040.
Point 2: Line 042: please change “Lithium” to “lithium”
Response 2: Thanks, we have corrected it in the revision on Line 042.
Point 3: Line 048: weight percent?
Response 3: Thanks, we have corrected it in the revision on Line 047-50. ‘’ÄŒerný et al. [3] reported that K2O+Na2O+Li2O+Rb2O+Cs2O(R2O) of beryl in Be-Nb-Ta type pegmatite was 0.5–1.0 wt%. In pegmatite with lithium minerals, beryl is characterized by high Na, Al, and low Cs contents, when Cs minerals appear in pegmatite, the R2O content of beryl is generally above 2% wt, especially the content of Cs2O is high [3].’’
Point 4: Line 049: reference for this information?
Response 4: Thanks, we have added cite in the revision on Line 050. [3] ÄŒerný, P.; Hawthorne, F.S. Refractive Indices Versus Alkali Contents in Beryl: General Limitation and Applications to Some Pegmatitic Types. Canadian Mineralogist. 1976, 14, 491–497.
Point 5: Line 054: please delete “On the other hand,”
Response 5: Thanks, we have deleted it in the revision (see Line 053).
Point 6: line057: please change “Ne” to “Nb” Also, most important compared to what?
Response 6: Thanks, we have corrected it in the revision on Line 056-58. The Dakalasu No.1 pegmatite is one of the few Be-Nb-Ta type pegmatites in the Altai, Xinjiang. There are only five Be-Nb-Ta type pegmatite deposits(Amusitai, Husite, Qukuer, Jiamukai, and Dakalasu) in Altai, Xinjiang(Zou and Li, 2006).
Point 7: Line 073: please change “discussed” to “inferred”
Response 7: Thanks, we have corrected it in the revision on Line 074.
Point 8: Line 092: what is a typical size of the pegmatite . . . width and length at surface?
Response 8: According to Zou and Li (2006), over 1000 pegmatites are distributed in the Dakalasu-Kekexi'er pegmatite field and most of them occur as dikes or lenticular bodies, and are 5 to 500 m long and from 0.5 to 30 m thick.
Point 9: Line 117: can you add a space at the beginning and end of each label in Figure 3? It would make each label much easier to read. Look at “Brl” in c and d, “Intermediate in 3b.
Response 9: Thanks, we have added a space at the beginning and end of each label in revised Figure 3 (see Line 118).
Point 10: Line 118-123: can you state the length of the “scale’ in photos b,c,d? For example, how long is the hammer in 3b and 3d? The pen length in 3c?
Response 10: The pen is about 13.5cm long, and the hammer is about 30cm long, we have added these on Line 125
Point 11: Line 122: is it only beryl and tourmaline in 3d? I am guessing the tourmaline is black in 3d, but is the white mineral microcline and the gray mineral quartz? It’s hard to tell. I would describe 3d different than “Beryl coexists with a great deal of tourmaline.” Maybe “Beryl coexisting with abundant tourmaline (black), microcline (white) and quartz (gray).”
Response 11: Thanks for pointing this out. In the revision, we have rewritten this sentence in the revision(see Figure 3d Line 122-123)
Point 12: Line 124: Figure 4 captions needs: 1) what does CGM stand for? 2) granulargranite is two words. 3) why the two symbols for quartz? 4) why the two symbols for microcline? What do each indicate? Do the different size symbols refer to crystal grain size. Please inform the reader
Response 12: Thanks for pointing these out. 1) CGM stands for columbite group minerals (see Line 114 ). 2) We have corrected it in the revision (see Figure. 4). 3),4) Different size symbols of quartz and microcline refer to crystal grain size and color. We have added them in the revision (see Line 108-109)
Point 13: Line 128-130: so this is the first time you are describing beryl I, beryl IIa and beryl IIb. How are each of the three beryl types defined? What makes them different?
Response 13: Thanks. In the revision, we have put the Beryl petrography content of Results into the chapter of Samples (see Line 130-144).
Point 14: Line 137: where was the EMPA work done?
Response 14: Thanks, we have added it in the revision (see Line 167 ).
Point 15: Line 144: at the top of table 1, what are the numbers in parentheses after the sample number? If they represent the number of analyses, then please provide some statistics on the variation of the data (maybe 1 sigma or 2 sigma error bars on each oxide). Same thing regarding table 2.
Response 15: Thanks, the numbers in parentheses represent the number of analyses. Statistic data are in the supplemental material (Table. S1)
Point 16: Line 147: please change “form” to “from”
Response 16: Thanks, we have corrected it in the revision (see Line 176)
Point 17: Line 166: resolution of what?
Response 17: Thanks, we have corrected it in the revision (see Line 194).
(resolution is 4.8 cm−1). This is the resolution of the Spectroscope
Point 18: Line 170: please change “Mirothermometric” to “Microthermometric”
Response 18: Thanks, we have corrected it in the revision (see Line 253)
Point 19: Line 174-179: Instead of describing the phases encountered while heating and cooling, ca you just tell the readers what phases are present at room temperature? And what kind of variation do you see at room temperature? I see where you describe the inclusion types later on, but it belongs up front.
Response 19: Thanks, for your suggestions. We have moved the contents of the previous Chapter 4.2 to Chapter 3.5. (see Line 198-227).
Point 20: Line 180-183: are the estimated phase ratios used later on?
Response 20: Yes, it is an important parameter to calculate ,, and in the inclusions. (X=mole fraction)
Point 21: Line 186-187: please refer to them as “final Ice melting temperatures” and “final clathrate dissolution temperatures.” Recall that both ice and clathrate are melting until they disappear.
Response 21: Thanks, we have corrected it in the revision (see Line 260-262)
Point 22: Line 193-195: please describe the homogenization behavior of all the inclusion fluids. Homogenization to liquid? Homogenization to vapor? It sounds like the latter.
Response 22: Thanks for pointing this out. Homogenization to vapor is right. We have corrected it in the revision (see Line 267-268)
Point 23: Line 200-213: To the reader, these are not Results . . . instead these are descriptions of the beryl in hand specimen. They belong up front.
Response 23: Thanks, we have moved this section to Chapter 3. 3.1 Sample (See Line 130)
Point 24: Line 215-219: Can you please take better photos of Fig. 5c and 5d. Nearly half of each is covered in a shadow!
Response 24: Thanks, we have taken better photos of Figure 5 in the revision (see Line 145)
Point 25: Line 252-253: May I ask how do you know these are melt inclusions? The Type B inclusions could be multiple-solid inclusions that were not trapped as melt? What evidence to you have that these are melt inclusions?
Response 25: Thanks for pointing this out. Raman Spectroscopy shows the type B inclusions couldn’t be multiple-solid inclusion, and the inclusion petrography is similar to Thomas et al (2009) observation to type B melt inclusion.
1.
2.
3.
4.
5.
Fig .3b A crystallized volatile-rich type-B melt inclusion in the (0001) plane. The arrows outline the hexagonal shadow(Thomas et al., 2009)
Point 26: Line 255: please change “inclusion” to “inclusions” also “is” to “are”
Response 26: Thanks, we have corrected it in the revision (see Line 207)
Point 27: Line 256: please change “inclusions” to “inclusion”
Response 27: Thanks, we have corrected it in the revision (see Line 209)
Point 28: Line 257-260: This part needs help. You must describe the criteria for inferring primary and secondary inclusions here. Negative crystal shape versus irregular shape isn’t good enough. More information about how the inclusions occur in the quartz and beryl is needed. You must convince the reader that you are investigating inclusions that are directly associated with the formation of beryl and quartz. See Roedder’s 1984 criteria. Not sure if studying the secondary inclusions is even necessary for this paper. See below on Table 3 and Figure 9.
Response 28: The discrimination of primary and secondary inclusions in beryl and quartz follow Rodder’s 1984 criteria. The EMS 2 Fig. S1 shows the fluid inclusion assemblage (FIA) for judging inclusions. The inclusions are parallel to the c-axis of beryl, which is considered primary inclusions. We have deleted the data of secondary inclusions of the new revision.
Point 29: This is applicable for line 261 too!
Response 29: See Response 28.
Point 30: Line 270-274: do you know, based on Raman spectra, what the daughter silicates are?
Response 30: The daughter silicates are quartz, based on Raman spectra.
Point 31: Line 254-274: you describe the 4 types of fluid inclusions, but you don’t write anything here about whether they constitute a fluid inclusion array (FIA) or fluid inclusion assemblage (FIA). This is important because if types 1,2, and 3 are found in the same FIA, then one could argue for fluid immiscibility. But, if they are not part of the same array, they could have been trapped at different times. This is a significant detail !!!!
Response 31: Thanks for your suggestions. I have put the figure used for this point in the new EMS 2 Fig. S1.
Point 32: Line 279: You state type A, but types A and B are shown in “a.” Also, delete the first A, you just need to write “(a) Types A and B inclusions . . . .”
Response 32: Thanks, we have corrected it in the revision (see Line 230)
Point 33: Line 280: You state type B, but type A is shown in “b.” Again, delete the first A, you just need to write “(b) Type B water poor . . . .”
Response 33: Thanks, we have corrected it in the revision (see Line 231)
Point 34: Line 281: a type A melt inclusion is show in image (c) rather than Type B as stated.
Response 34: Thanks, we have corrected it in the revision (see Line 232)
Point 35: Line 281: are these two inclusion types part of the same array? Is it possible the two type 1 inclusions have some CO2 and you just can’t see the two bubbles because they are not flat like the two inclusions above?
Response 35: Thanks for pointing these out. In addition to the petrography of inclusion, the phase transformation of microthermometry is also the basis for judging the type of inclusions.
Point 36: Line 282: it seems that the Type 2 inclusions in image E are much more CO2 rich than those in image d. It appears that necking occurred after phase separation to yield the CO2-rich portions. The Type 1 inclusions in image E between the two dashed lines are very small, so how can you determine that no CO2 is present?
Response 36: see Response 35.
Point 37: Line 283: In image F, how do you know the crystallographically controlled inclusions are fluid inclusions? Hard to tell just from images. Do you observe both the liquid and vapor CO2? I am guessing that the inclusions are parallel to the c-axis of beryl, if so, you should state this. A close-up image of one of these inclusions would help significantly.
Response 37: Thanks for your suggestions. In the revision, we have added the sentence ’’….. are parallel to the c-axis of beryl’’.(see Line 234) and we have added a close-up image of one of these inclusions in the supplement.
Point 38: Line 284. What is the identification of mineral or minerals in figure G? See below:
Response 38: See response 30.
Point 39: Line 288-289: How do you know this is glass? Glass has a very specific Raman spectrum and I don’t see any evidence for this. Also, you state the inclusion in 7a and 8a is from a beryl host, yet your Raman spectrum shows a quartz peak. Is this a quartz grain in the melt inclusion? Is it really coming from a quartz host instead? There are no background beryl peaks? None of this makes sense, please clarify this! I would make the same comment about the quartz peak in 8b. That is a pretty significant CH4 peak in the melt inclusion vapor!.
Response 39: Thanks for your suggestion. Figure 8a isn’t a Raman spectrum of glass. It is not that there are no background beryl peaks. But that is some parts of the inclusion, the background of quartz or water is too intensive, which leads to the unobvious background peaks of beryl. see response 25 figure. Yes, there is indeed some vapor in inclusions containing CH4.
Point 40: Line 293-297: What do you mean by partial homogenization temperature? I’ve never heard such a term. Do you mean decrepitation temperature? Is this the temperature at which the inclusion exploded and leaked its constituents? Some have had success heating these inclusions in the diamond anvil cell (Schmidt et al 1999, Darling and Bassett, 2002).
Response 40: Thanks for pointing it out. Partial homogenization temperature of type 4 inclusions means liquid phase homogenization in type 4 inclusions. It is very difficult to determine the total homogenization temperature of the type 4 inclusions. In order to be avoided confusion with the previous term(Schmidt et al 1999, Darling and Bassett, 2002), we have deleted it in the revision on Line 293-294.
Point 41: Line 299: Table 3 lists about 370°C not 350°C.
Response 41: Thanks, we have corrected Table.3 in the vision
Point 42: Line 300: The reference for determining salinity from final ice melting temperatures is not [38]. The Darling (1991) reference is for final clathrate melting temperatures in the presence of both CO2 liq and CO2 vapor. I would use Bodnar (1993) for final ice melting temperatures
Response 42: Thanks, we have corrected it in the vision on reference [38].
Point 43: Line 304: where is “275°C and 450°C, with a distinctive peak around 325 to 350°C” listed in Table 3?
Response 43: Thanks, we have deleted it in the vision (see Line 310-311)
Point 44: Line 305: Are these really “final CO2 melting temperatures,” maybe, if they are that low. If close to -56.6°C, they are CO2 triple point temperatures, and if they are between -66.3°C and -57.3°C, then there is a significant quantity of something other than CO2 in the vapor bubble. Maybe it’s CH4.
Response 44: Thanks, The vapor bubble in fluid inclusion contain CO2 and CH4 based on Raman spectrum. The final CO2 melting temperatures is mostly concentrated around -60℃.
c
Point 45: Line 305-307: See below, but these are “final clathrate melting temperatures.” And the reference here is [38].
Response 45: Thanks for your suggestion, we have put the [39] reference at the end of line 307.
Point 46: Line 307-308: WHAT? All of them show CO2 homogenization by critical behavior? None show homogenization to liquid? None show homogenization to vapor? Wow, that is remarkable. To what do you attribute this very unusual behavior? Also, 15°C and 28.4°C is NOT a narrow range! If they are pure CO2, they will homogenize by critical behavior at +31.1 C only if they have the critical density. Less if there is some CH4 or another gas in there. Is this behavior really observed?
Response 46: Thanks for pointing this out. We have used the wrong term in the original paper,and we corrected it in the revision ‘’… the homogenization of CO2 (to liquid CO2) ….’’ see Line 307 and Line 317 .
Point 47: Lines 309-314: List the temperatures of phase transformations from lowest to highest, ending with total homogenization temperatures. Also, describe the homogenization behavior . . . are the inclusions homogenizing by vapor bubble disappearance or by vapor bubble expansion? Same thing with the CO2 homogenization, vapor bubble disappearance or by vapor bubble expansion? The CO2 triple point temperatures indicate the presence of a gas other than CO2, please state this! How does this affect salinity estimates? Put the [38] reference at the end of line 313, if that is where the calculation comes from.
Response 47: we have corrected it in the revision on Line 303-320 . Thanks for your suggestion, we have put the [39] reference at the end of line 313.
Point 48: Table 3: Why are the fluid inclusion TYPES not listed in Table 3?
Response 48: Thanks for pointing this out. In the revision, we have added FI types in Table 3.
Point 49: Table 3 abbreviations: Tm, ice is the “final” ice melting temperature, and Tm, clath is the “final” clathrate melting temperature. Both are melting along their respective liquid until they disappear completely.
Response 49: Thanks, we have corrected it in the revision (see Table 3).
Point 50: Table 3 – I would place the negative sign in front of the actual temperatures of final ice melting rather than above in the heading. Very confusing.
Response 50: Thanks, we have corrected it in the revision (see Table 3).
Point 51: Table 3 and Figure 9: Why are you studying secondary fluid inclusions? Why not just focus on what the primary inclusions are telling you about the formation of quartz and beryl? If I were to author this paper, I would focus on providing good descriptions of why the inclusions are primary in origin and then only write about the data from the primary inclusions. What are the secondary inclusions telling us? . . . about a fluid that was trapped after the quartz and beryl formed? How useful is that to us?
Response 51: Thanks for your suggestion, we have deleted the content about secondary inclusions in the revision.
Point 52: Line 352: “In terms of trace elements.” is not a sentence.
Response 52: Thanks, we have deleted it in the revision (see Line 355)
Point 53: Line 376: Figure 10: You plot all of the Na, Cs and Fe data from Table 1, but they are not listed in Table 1, just the averages are. If this data are in the supplemental material then please state it here.
Response 53: Thanks, these data are in the supplemental material (Table. S1)
Point 54: Line 382-384: This is unclear. These pegmatites don’t have silicate melt inclusions in them? Please state this more clearly.
Response 54: Thanks, we have corrected it in the revision (see Line 386-391)
Point 55: Line 386-387: are you saying the other pegmatites have more complex array of minerals in them, indicating a higher degree of evolution? It’s not clear.
Response 55: The daughter minerals of melt-fluid inclusions in Dakalasu No.1 are mostly quartz, and the number of daughter minerals is single. The structure of inclusions is far less complex than that of melt-fluid inclusions in other deposits. Considering that Li mineralization is widely developed in other deposits such as Koktokay No.3 pegmatite, the evolution degree of Li mineralization is certainly higher than that of Be mineralization.
Fig. Melt-fluid photographs of pegmatite in Koktokay No.3 pegmatite, Xinjiang (Lu et al., 2004)
Point 56: Line 391: “inferred” rather than “judged”
Response 56: Thanks, we have corrected it in the revision (see Line 394-395)
Point 57: Line 393: “suggesting” rather than “possibly proving” We don’t prove theories in science. We find only supporting evidence for or against.
Response 57: Thanks, we have corrected it in the revision (see Line 397)
Point 58: Line 398: but you don’t say how the CO2 homogenized (to liquid or vapor). This is significant if we are to estimate the bulk density of the inclusions. And, again that is not a narrow range.
Response 58: Thanks, we have corrected it in the revision (see Line 401)
Point 59: Line 399: Again, these should be showing “CO2 triple point temperatures” especially if they are anywhere near -56.6°C.
Response 59: Thanks, we have corrected it in the revision (see Line 401-402)
Point 60: Line 400: Again, the degree of fill is useful only if we know the density of the CO2 which we cannot determine because you don’t describe the homogenization behavior.
Response 60: Thanks, we have added the homogenization behavior of CO2 in the revision (see Line 402-404)
Point 61: Line 419-421: Figure 12 is incompletely described. What are the curved lines? What are the two numbers in parenthesis? Are they pressure on the H2O-CO2-NaCl solvus in kilobars and temperature in centigrade?
Response 61: Thanks, we have added it in the revision (see Figure.12)
Point 62: Line 403-404: Most importantly, are you inferring the temperature and pressure of trapping from this plot? Gosh, I hope not. Are you inferring the fluids formed on the side of the H2O-CO2-NaCl solvus? If so, what fluid inclusion evidence do you have to support the trapping of immiscible liquids? If there is no evidence of fluid immiscibility then all you can infer is that the fluids formed above the location of the H2O-CO2-NaCl solvus, not on it! This plot does nothing except to show the range of fluid compositions, and to say the fluids were trapped above the solvus.
Response 62: Thanks for your suggestions. Figure.12 is not evidence of fluid immiscibility, it is only used to infer the trapping temperature and pressure condition of CO2 inclusions (Sirbescu et al. 2003, Li, 2006). The basis for judging the existence of fluid immiscibility in beryl and quartz comes from petrographic observation of inclusions.
Different types of fluid inclusions with similar gas-liquid ratios can always be observed in the same array of FIA along the beryl c-axis. Combined with their total homogenization temperature and salinity, the inclusions may have trapped the fluid further separating at the time.
In the next step, we plan to carry out LA-ICPMS for single inclusion to improve the study of fluid immiscibility and element fractionation mechanism.
Point 63: Line 405: I am guessing you mean “crystallized from the melt” not “separated from the melt”
Response 63: Thanks, we have corrected it in the revision (see Line 408)
Point 64: Line 409-410: But what is your evidence that the fluid further experience unmixing? In theory, what you are stating could happen, but what evidence to you have that it did? I don’t see any evidence of fluid immiscibility in Figure 12?
Response 64: See response 62.
Point 65: Line 433-437: I have no idea what is meant by the term “granite bedding.” Please explain.
Response 65: Thanks for pointing this out. The term “granite bedding” means layered sheeting structure. The Dakalasu pegmatite dikes occur mostly with granite bedding (sheeting structure) (Fig. b).
Figure. Field photographs of Dakalasu
Context of bedding-parallel faults (BPFs) and sampling in Trescl´eoux (a), Espr´eaux (b) and Saint-Didier (c). The normal faults (NF) offsetting the limestone and clay layers and the bedding-parallel faults (BPFs) are indicated. The limestone and clay layers are represented in light and dark grey, respectively. The numbering indicates the location of the BPF sampling for the thin sections. Examples of details of sampling are presented in the right panel(Lemonnier et al., 2020).
Detailed lithostratigraphic column of the Khanguet Aicha cross-section with lithological description and the bedding parallel veins (BPV) distribution along the Bouhedma Formation. The sampled BPV are highlighted by underlining BPV numbers (from KA01 to KA46)(Abaab et al., 2021).
Point 66: Line 463: Please replace “the reasons for” with “inferred to be the cause of”
Response 66: Thanks, we have corrected it in the revision (see Line 468).
Renfence
- Thomas, R.; Davidson, P.; Badanina, E. A melt and fluid inclusion assemblage in beryl from pegmatite in the Orlovka amazonite granite, East Transbaikalia, Russia: implications for pegmatite-forming melt systems. Mineral Petrol. 2009a, 96, 129–140. 10.1007/s00710-009-0053-6.
- Lu, H.Z.; Fan, H.R.; Ni, P.; Ou, G.X.; Chen, K.; Zhang, W.H. Fluid Inclusion. Thermodynamics. Science Press (in Chinese). 2004; pp. 291.
- Sirbescu, M.C.; Nabelek, P.I. Crystallization conditions and evolution of magmatic fluids in the Harney Peak Granite and associated pegmatites, Black Hills, South Dakota—Evidence from fluid inclusions. Geochem. Cosmochim. Acta. 2003, 67, 2443–2465. 10.1016/s0016-7037(02)01408-4.
- N, L.; C, H.; V, R.; M, R.; M, B.A.; J, S. Microstructures of bedding-parallel faults under multistage deformation: Examples from the Southeast Basin of France. Journal of Structural Geology. 2002. 140, 104-138. https://doi.org/10.1016/j.jsg.2020.104138.
- Abaab, N.; Zanella, A.; Akrout, D.; Mourgues, R.; Montacer, M. Timing and distribution of bedding-parallel veins, in evaporitic rocks, Bouhedma Formation, Northern Chotts, Tunisia. Journal of Structural Geology. 2021 153, 1044-61. https://doi.org/10.1016/j.jsg.2021.104461.
- Li, J.K. Mineralizing Mechanism and Continental Geodynamics of Typical Pegmatite Deposits in Western Sichuan, China. Ph.D. thesis, China University of Geosciences, Beijing, 2006.

Reviewer 3 Report
r16-17 - “textural and chemical compositions” sounds odd, “textural relationships and…” suggested instead
r57 Be-Nb-Ta
r67 Initial capital letter
r75-76 Actually it is adjacent to eastern Kazakhstan, southern Russia and western Mongolia, or Kazakhstan to the west, Russia to the north, aso. - a matter of relativity.
r80 namely or included?
r82 Misspelled: Faults(two times), Tarim (in inset 1a); Kekexier (or Kekexi’er ?) - capital letter. The name of the terranes is confusing because what is named North Altai, Northwest Altai, Central Altai are all three located in the south or south-west (Chinese) Altai. The reasons for separating III from IV are not obvious and have to be explained.
r83 CAOB abbreviation not explained
r85 Wouldn’t be better to cite https://doi.org/10.1016/j.oregeorev.2018.02.022 instead of [68] ?
r106 why inward?
r109 Tourmaline is arranged along the contact with (the) biotite granite
r110-111 why “the excessive position” ? Please rephrase.
r112 are they giant indeed?
r122 abundant tourmaline?
r115-116 Seems inconsistent with fig. 4, where large microcline is represented in the core zone.
r138-139 A beam with a diameter of 2 µm is not focused. A slightly defocused beam?
r146 Sample Beryl 1 does not fulfil the condition Be=3-Li. Typing mistake or other calculation method?
r160 Abbreviation 1RSD not explained
r202 & following: shallow=pale, light ?
r 203 Extra comma.
r 212 unzoned
r225-226 Unclear, is it a pseudomorph after beryl?, an aggregate replacing the central zone of a crystal?
r271 Meaning of “sub-minerals” unclear
r275 Monophase is not exactly a fortunate characterization, considering that they are part of a polyphase inclusion. No need to stress this feature of daughter minerals, the normal case is that they precipitate as single crystals.
r288 probably aqueous glass - aqueous means rich in water
r287 CO2 in b
r295 It is very difficult to determine the homogenization temperature of type 4 inclusions… - a sequel of this assertion should be given to justify why was it mentioned.
r309-310 a liquid-vapor two-phase composition (or something similar), carbon-rich three-phase inclusions
r318 Mdn=median
r337 Sentence missing a predicate.
r 350 showing
r 352 Isolated “In terms of trace elements”
r359 exclusive fluid - unclear meaning
r363 Fluorine-rich
r 368 the mechanism by which… might be…
r 374 [TiF6]2-
r412 will coexist
r414 1; the salinity range…
r457 …to form large and euhedral beryl crystals…
r492-493 Lenghty web address better omitted because the publication, volume, page are already given and the paper has no doi.
r577 capital letters for toponyms (Velasco, Pampanea)
r619 Günther
r620 Mole
Author Response
Please see the attachment
Response to Reviewer 3 Comments
Point 1: r16-17 - “textural and chemical compositions” sounds odd, “textural relationships and…” suggested instead
Response 1: Thanks, we have used “textural relationships and…” in the revision on Line 17.
Point 2: r57 Be-Nb-Ta
Response 2: Thanks, we have corrected it in the revision on Line 56.
Point 3: r67 Initial capital letter
Response 3: Thanks, we have corrected it in the revision on Line 64.
Point 4: r75-76 Actually it is adjacent to eastern Kazakhstan, southern Russia, and western Mongolia, or Kazakhstan to the west, Russia to the north, also. - a matter of relativity.
Response 4: Thanks, we have corrected it in the revision on Line 76-77.
Point 5: r80 namely or included?
Response 5: Thanks, we have corrected it in the revision on Line 81. The Chinese Altai orogenic belt is divided into five terranes, namely the North Altai, Northwest Altai, Central Altai, Qiongkuer-Abagong, and South Altai.
Point 6: r82 Misspelled: Faults(two times), Tarim (in inset 1a); Kekexier (or Kekexi’er ?) - capital letter. The name of the terranes is confusing because what is named North Altai, Northwest Altai, Central Altai are all three located in the south or south-west (Chinese) Altai. The reasons for separating III from IV are not obvious and have to be explained.
Response 6: Thanks, we have corrected misspelled words in the revision on Line 83-86 (Figure 1).
North Altai, Northwest Altai, and Central Altai are divided according to the faults of the Altai orogenic belt(Fig. S1) (Windley et al., 2002).
Central Altai (III). This terrane forms the central part of the Altai orogen in China. It contains widespread high-grade metamorphic rocks and abundant granites, some Neoproterozoic to Silurian metasediments but no island arcs.
Qiongkuer-Abagong (IV). This terrane contains two formations: the Kangbutiebao Formation and the Altai Formation. The predominant Kangbutiebao Formation consists of upper Silurian to lower Devonian arc-type volcanic and pyroclastic rocks that are 1–2 km thick and minor basic volcanic rocks and spilites. In the far eastern Altai, this formation consists mostly of meta-andesite (Windley et al., 2002,2007; Xiao et al., 2004).
Figure. Geological map of the Chinese Altai showing main terranes referred to by number in the text.
Point 7: r83 CAOB abbreviation not explained
Response 7: Thanks, we abbreviated the Chinese Altai orogenic belt when it first appeared on Line 76.
Point 8: r85 Wouldn’t be better to cite https://doi.org/10.1016/j.oregeorev.2018.02.022 instead of [68]?
Response 8: Thanks for your suggestions. Thanks, we have corrected it in the revision on Line 86.
Point 9: r106 why inward?
Response 9: Thanks for pointing this out, this sentence has been rewritten in the revision.
According to the previous studies [14,15,17], the Dakalasu No. 1 pegmatite can be divided into four zones (Fig. 4), from the contact with Dakalasu biotite granite inwards, are the wall zone, intermediate zone, replacement zone, and core zone(see Line 106-108).
When introducing the internal zonation of pegmatite, it usually starts from contact with granite or wall rock and then moves inward into the border zone, wall zone, intermediate zone, and core zone. Most of the terms that are in use today to describe the internal zonation were established by Cameron et al. (1949). These are reviewed by London (2008, 2014a).
Point 10: r109 Tourmaline is arranged along the contact with (the) biotite granite
Response 10: Thanks, we have corrected it in the revision on Line 110-111.
Point 11: r110-111 why “the excessive position” ? Please rephrase.
Response 11: Thanks, we have corrected it in the revision on Line 111.
Modify: Garnet is developed in the transition area between the wall zone and intermediate zone
Point 12: r112 are they giant indeed?
Response 12: Thanks, they are giant indeed. According to Feng et al. (2020,2021), we redefined the size of muscovite as megacrystic on Line 113.
Point 13: r122 abundant tourmaline?
Response 13: Thanks, we have corrected it in the revision on Line 123.
Point 14: r115-116 Seems inconsistent with fig. 4, where large microcline is represented in the core zone.
Response 14: Thanks for pointing this out. In the lower right corner of Figure 3b, quartz and massive microcline was observed in the field near the core zone, and line existed symbiosis in the core zone(20DK-1).
|
|
Point 15: r138-139 A beam with a diameter of 2 µm is not focused. A slightly defocused beam?
Response 15: Thanks, we have rewritten this sentence in the revision (see Line 168). Major element compositions of beryl were measured using a JEOL JXA-8100 and CAMECA SXFive electron microprobe operated in wavelength dispersive spectrometer mode at IGGCAS, respectively. EPMA under the following operation conditions: an accelerating voltage of 15 kV, a beam current of 20 nA with beam diameters of 2 μm, and a 10–30 s peak counting time.
Point 16: r146 Sample Beryl 1 does not fulfil the condition Be=3-Li. Typing mistake or other calculation method?
Response 16: Thank you very much for your professional review! Sample beryl 1 does not fulfi the condition Be= 3-Li, which is my calculation error. We have corrected the mistake in the revision on Table 1 (atoms: Be=2.75, Li=0.25, obtained by Table S1).
Point 17: r160 Abbreviation 1RSD not explained
Response 17: Thanks, we have added it in the revision on Line 189 (relative standard deviation=RSD).
Point 18: r202 & following: shallow=pale, light ?
Response 18: Yes, that’s right. ‘’shallow’’ has been changed into ‘’pale’’ in the paper. (see on Line 133)
Point 19: r 203 Extra comma.
Response 19: Done
Point 20: r 212 unzoned
Response 20: Thanks, we have corrected it in the revision on Line 149.
Point 21: r225-226 Unclear, is it a pseudomorph after beryl? an aggregate replacing the central zone of a crystal?
Response 21: It is a pseudomorph after beryl, an aggregate replacing the central zone of a crystal.
Dakalasu No.1 pegmatite
Point 22: r271 Meaning of “sub-minerals” unclear
Response 22: Thanks for pointing this out. We have replaced “sub-mineral” with “daughter mineral” in revision.
Point 23: r275 Monophase is not exactly a fortunate characterization, considering that they are part of a polyphase inclusion. No need to stress this feature of daughter minerals, the normal case is that they precipitate as single crystals
Response 23: Thanks, We have deleted it in the revision(see Line 226).
Point 24: r288 probably aqueous glass - aqueous means rich in water
Response 24: Done
Point 25: r287 CO2 in b
Response 25: Done
Point 26: r295 It is very difficult to determine the homogenization temperature of type 4 inclusions… - a sequel of this assertion should be given to justify why was it mentioned.
Response 26: Thanks for pointing it out. Partial homogenization temperature of type 4 inclusions means liquid phase homogenization in type 4 inclusions. It is very difficult to determine the total homogenization temperature of the type 4 inclusions. In order to be avoided confusion with the previous term(Schmidt et al 1999, Darling and Bassett, 2002), we have deleted it in the revision on Line 293-294.
Point 27: r309-310 a liquid-vapor two-phase composition (or something similar), carbon-rich three-phase inclusions
Response 27: Thanks, we have corrected it in the revision on Line 312-313.
Point 28: r318 Mdn=median
Response 28: Thanks, we have corrected it in the revision on Table 3.
Point 29: r337 Sentence missing a predicate.
Response 29: Beryl I occur in the wall zone, beryl IIa, and IIb occur in the intermediate zone of the Dakalasu No.1 pegmatite. See Line 340.
Point 30: r 352 Isolated “In terms of trace elements”
Response 30: Thanks, we have deleted it in the revision(see Line 355)
Point 31: r 350 showing
Response 31: Thanks, we have corrected it in the revision on Line 353.
Point 32: r359 exclusive fluid - unclear meaning
Response 32: Thanks for pointing this out, we have used the wrong term about ‘’exclusive fluid’’.
What we want to express is ‘’exsolving fluid’’ in the sentence, we have corrected it in the revision on Line 362.
Point 33: r363 Fluorine-rich
Response 33: Thanks, we have corrected it in the revision on Line 366.
Point 34: r 368 the mechanism by which… might be…
Response 34: Thanks, we have corrected it in the revision on Line 371-372.
Point 35: r 374 [TiF6]2-
Response 35: Thanks, we have corrected it in the revision on Line 374.
Point 36: r412 will coexist
Response 36: Thanks, we have corrected it in the revision on Line 414.
Point 37: r414 1; the salinity range…
Response 37: Thanks, we have corrected it in the revision on Line 416.
Point 38: r457…to form large and euhedral beryl crystals…
Response 38: Thanks, we have corrected it in the revision on Line 461.
Point 39: Lenghty web address better omitted because the publication, volume, page are already given and the paper has no doi.
Response 39: Done
Point 40: r577 capital letters for toponyms (Velasco, Pampanea)
Response 40: Thanks, we have corrected it in the revision on Line 584 .
Point 41: r619 Günther
Response 41: Thanks, we have corrected it in the revision on Line 626.
Point 42: r620 Mole
Response 42: Thanks, we have corrected it in the revision on Line 627.
Reference
- Windley, B.F.; Kröner, A.; Guo, J.; Qu, G.; Li, Y.; Zhang, C.; Neoproterozoic to Paleozoic geology of the Altaid orogen, NW China: new zircon age data and tectonic evolution. Journal of Geology. 2002, 110, 719–737. https://doi.org/10.1086/342866.
- Windley, B.F.; Alexeiev, D.; Xiao, W.J.; Kröner, A.; Badarch, G. Tectonic models for accretion of the Central Asian Orogenic Belt. Journal of the Geological Society London. 2007, 164, 31–47. 10.1144/0016-76492006-022.
- Xiao, W.J.; Windley, B.F.; Badarch, G.; Sun, S.; Li, J.L.; Qin, K.Z.; Wang, Z. Palaeozoic accretionary and convergent tectonics of the southern Altaids: implications for the growth of Central Asia. Journal of the Geological Society London. 2004, 161, 339–342. 10.1144/0016-764903-165.
- Cameron, E.N.; Jahns, R.H.; McNair, A.H.; Page, L.R. Internal structure of granitic Econ. Geol. Mono. 1949, 2, 115.
- London, D. Pegmatites. Canadian Mineralogist Special Publication. 2008, 10, pp. 368.
- London, D. A petrologic assessment of internal zonation in granitic pegmatites. Lithos. 2014a, 184–187, 74–104.
- Feng, Y.G.; Liang, T.; Linnen, R.; Zhang, Z.; Zhou, Y.; Zhang, Z.L.; Gao, J.G. LA-ICP-MS dating of high-uranium columbite from no.1 pegmatite at Dakalasu, the Chinese Altay orogen: Assessing effect of metamictization on age concordance. Lithos. 2020, 362–363, 1054–1061. https://doi.org/10.1016/j.lithos.2020.105461.
- Feng, Y.G.; Liang, T.; Lei, R.X.; Ju, M.H.; Zhang, Z.L.; Gao, J.G.; Zhou, Y.; Wu, C.Z. Relationship between undercooling and emplacement of rare-element pegmatites-Thinking based on field observations and pegmatite geochronology. Journal of Earth Science and Environment. 2021, 43. (in Chinese).

Round 2
Reviewer 2 Report
The authors have made significant changes to the text and I think, based on their response to my comments, it is now publishable.
I would like to make clear, however, in the author's response 62, that the homogenization of the CO2-bearing inclusions reflect a point on the H2O-CO2-NaCl solvus, but the fluid could have been trapped anywhere along an isochore above the solvus. Unless one has direct evidence of fluid immiscibility, the total homogenization temperatures (and inferred pressures) are not the temperature and pressure of trapping. They are only minimum temperatures and pressures of trapping.
Lastly, regarding fluid immiscibility . . . the best evidence for fluid immiscibility are fluid inclusions with different phase ratios and homogenization behaviors (i.e. some by vapor bubble expansion, others by vapor bubble shrinkage) WITHIN the same FIA but having similar homogenization temperatures.
The authors state "Different types of fluid inclusions with similar gas-liquid ratios can always be observed in the same array of FIA along the beryl c-axis. Combined with their total homogenization temperature and salinity, the inclusions may have trapped the fluid further separating at the time." I have no issue if the authors are referring to various fluids separating from the melt, but I have an issue if the authors are referring to aqueous fluid unmixing.
I would just have this clarified in the text . . . otherwise good to go . . .
Author Response
Dear Reviewer. Thanks for your thorough reviews and insightful suggestions on the original vision of our manuscript. Re-reading article of Frantz et al. 1992 and Sirbescu et al. 2003, and re-understand the meaning of the phase diagram. Figure 12 can only reflect the fluid could have been trapped anywhere along an isochore above the solvus, and can only represent the minimum temperature and pressure at which the fluid is trapped. We have corrected it in the 2nd revised version on line 406 "The minimum trapped pressure (temperature) of CO2-NaCl-H2O inclusions of beryl and quartz is 200 Mpa (500°C)." Thank you again for mentioning the criteria for fluid immiscibility.
About the statement of us "Different types of fluid inclusions with similar gas-liquid ratios can always be observed in the same array of FIA along the beryl c-axis. Combined with their total homogenization temperature and salinity, the inclusions may have trapped the fluid further separating at the time." We are indeed referring to various fluids separating from the melt.
Reference
- Frantz, J D.; Popp, R. K.; Hoering, T.C. The compositional limits of fluid immiscibility in the system H2O-NaCl-CO2 as determined with the use of synthetic fluid inclusions in conjunction with mass spectroscopy. Chem. Geol. 1992, 98, 237–255.
- Sirbescu, M.C.; Nabelek, P.I. Crystallization conditions and evolution of magmatic fluids in the Harney Peak Granite and associated pegmatites, Black Hills, South Dakota—Evidence from fluid inclusions. Geochem. Cosmochim. Acta. 2003, 67, 2443–2465. 10.1016/s0016-7037(02)01408-4.
- Liu, B. Fluid Inclusion Thermodynamics. Geological Publishing House (in Chinese).2000; pp. 252.
